# Research on MICP Restoration Technology for Earthen City Walls Damaged by Primary Vegetation Capping in China

**DOI:** 10.3390/microorganisms13122802

**Published:** 2025-12-09

**Authors:** Ruihua Shang, Chenyang Li, Xiaoju Yang, Pengju Han, Weiwei Liu

**Affiliations:** 1College of Architecture and Art, Taiyuan University of Technology, Taiyuan 030024, China; 2China-Central Asia “the Belt and Road” Joint Laboratory on Human and Environment Research, Key Laboratory of Cultural Heritage Research and Conservation, School of Culture Heritage, Northwest University, Xi’an 710127, China; 3National Research Center for Conservation of Ancient Wall Paintings and Earthen Sites, Dunhuang Academy, Dunhuang 736200, China; 4College of Civil Engineering, Taiyuan University of Technology, Taiyuan 030024, China; 5Taiyuan Institute of Cultural Relics and Archaeology, Taiyuan 030081, China

**Keywords:** *Sporosarcina pasteurii*, feature parameters, importance, growth characteristics, urease activity, the yield of calcite in CaCO_3_

## Abstract

As a typical representative of soft capping, primary vegetation capping has both protective and destructive effects on earthen city walls. Addressing its detrimental aspects constitutes the central challenge of this project. Because the integration of MICP technology with plants offered advantages, including soil solidification, erosion resistance, and resilience to dry–wet cycles and freeze–thaw cycles, the application of MICP technology to root–soil composites was proposed as a potential solution. Employing a combined approach of RF-RFE-CV modeling and microscopic imaging on laboratory samples from the Western City Wall of the Jinyang Ancient City in Taiyuan, Shanxi Province, China, key factors and characteristics in the mineralization process of *Sporosarcina pasteurii* were quantified and observed systematically to define the optimal pathway for enhancing urease activity and calcite yield. The conclusions were as follows. The urease activity of *Sporosarcina pasteurii* was primarily regulated by three key parameters with bacterial concentration, pH value, and the intensity of urease activity, which required stage-specific dynamic control throughout the growth cycle. Bacterial concentration consistently emerged as a high-importance feature across multiple time points, with peak effectiveness observed at 24 h (1.127). pH value remained a highly influential parameter across several time points, exhibiting maximum impact at around 8 h (1.566). With the intensity of urease activity, pH exerted a pronounced influence during the early cultivation stage, whereas inoculation volume gained increasing importance after 12 h. To achieve maximum urease activity, the use of CASO AGAR Medium 220 and the following optimized culture conditions was recommended: an activation culture time of 27 h, an inoculation age of 16 h, an inoculation volume of 1%, a culture temperature of 32 °C, an initial pH of 8, and an oscillation speed of 170 r/min. Furthermore, to maximize the yield of CaCO_3_ in output and the yield of calcite in CaCO_3_, the following conditions and procedures were recommended: a ratio of urea concentration to Ca^2+^ concentration of 1 M:1.3 M, using the premix method of *Sporosarcina pasteurii*, quiescent reaction, undisturbed filtration, and drying at room-temperature in the shade environment.

## 1. Introduction

Unlike the stone relics in Europe and America, China has a large number of earthen sites, which are widely distributed in different environments such as arid, semi-arid, humid, and coastal hot and humid areas. Among them, earthen city walls are undergoing irreversible deterioration. Their protection is a world-class challenge. These sites face two main challenges: (i) continuous exposure to environmental fluctuations, which accelerates deterioration; and (ii) their large scale, which makes artificial control methods unfeasible. Field studies at several earthen city walls in Shanxi Province have revealed significant biological colonization and growth, giving rise to a wild cultural landscape. In the early years of earthen sites protection, conservation professionals regarded this phenomenon as a negative disease that was not conducive to the protection of cultural relics, and carried out artificial intervention measures, including vegetation removal, soil filling, and chemical root eradication. After the 20th century, with the evolution of the concept of cultural heritage protection, the relationship between plants and sites has advanced considerably, establishing it as a key area of interdisciplinary research. Lv [1], Lu [2], Riegl [3], and Brandi [4] confirmed that these plants had a positive role in the inheritance of value and showed good physical and chemical protection functions for earthen city walls. However, not all native vegetation is suitable for retention. Years of work experience have led conservation professionals to reach a consensus that herbaceous plants are suitable for retention and woody plants are mostly recommended for conservative observation or direct removal. The core problem with woody plants is that root splitting fissures open water channels in the walls, causing rainwater to seep into the interior of the soil along the root system, forming wet migration (Figure 1) disturbances. Wet migration is closely related to the salt migration, dry–wet cycles, and freeze–thaw cycles in the soil, which are key to the development of surface and internal diseases in the soil; in severe cases, it may even cause the collapse of earthen city walls. Zhu et al. [5], Liu et al. [6], Bu et al. [7], Zhang et al. [8], Boruah [9], and Varnitha et al. [10] proved the particularity of buildings that are cultural relics. It is imperative to establish a protection technology that does not damage the original appearance and can efficiently improve the physical and chemical properties of the surface of the target defect material.

Microbial-Induced Carbonate Precipitation (MICP) is widely observed in nature, where microbial mineralization is diverse. Wu et al. [11], Liu et al. [12], Wang et al. [13], Whiftin et al. [14], Al-Tabbaa et al. [15] and Simone et al. [16] proved that the cementitious materials precipitated by mineralization could be applied to stone, historical buildings, concrete, and sandy soil materials, featuring functions such as film protection, self-restoration, and environmental-friendly alternatives. In recent years, Daryono et al. [17], Deng et al. [18], Wang et al. [19], and Murugan et al. [20] started to explore the combined effect of MICP technology with plants from the perspective of soil synergistic reinforcement and environmental protection. Xu et al. [21] and Arias et al. [22] proved that during the metabolic processes, microorganisms could produce urease amidohydrolase (urease). Wang [23] proved the microorganisms with urease catalysis could hydrolyze urea at a rate 10^14^ times higher than that without urease catalysis. Torres-Aravena et al. [24] showed the urease catalysis steps with urea hydrolysis, chemical equilibrium, heterogeneous nucleation, and continuous stratification. Studies confirmed that the combination of MICP’s biocementation with plant growth had the following advantages. (a) MICP solidified sandy soil for plant cultivation, and Chen et al. [25] proved each square meter of shrub could absorb 253.1 g of CO_2_ per year. He et al. [26] proved it was mainly used for the remediation of heavy metal-contaminated soil in the mining industry. (b) Liu et al. [27], Liu et al. [28], and Wang et al. [29] showed the CaCO_3_ crystals generated by the MICP process contributed in a dominant fashion to the soil crack suppression by filling crack space, bonding soil particles, and reducing soil water evaporation. (c) Liang et al. [30], Payan et al. [31], and Lopes et al. [32] showed the terms of resistance to erosion, and Zheng et al. [33] indicated that the cohesion and internal friction angle of the root–soil complex increased by approximately 400% and 120%, respectively. Calcium carbonate formed by MICP treatment had cementitious properties, which increased the cohesion and internal friction angle of the root–soil composite by about 400% and 120%, respectively. Gong et al. [34] and Zhu et al. [35] showed that the soil strength was significantly increased by 22.22%, the disintegration mass decreased by 11.36%, and the mass loss caused by rain erosion reduced by 88.55%. It had unique advantages in resisting dry–wet cycles and freeze–thaw cycles.

In this paper, *Sporosarcina pasteurii* (abbreviated hereafter as *S. pasteurii*)—produced by Leibniz Institute DSMZ in Germany and numbered DSM 33—is selected as the MICP target strain. Tang et al. [36] and Wu et al. [37] illustrated that it was extracted from soil and later found in many places such as limestone. Kang et al. [38] and Wong et al. [39] pointed out that it was an aerobic urease-producing bacterium with high degradation capacity, and it exhibited the advantages of non-toxicity, environmental friendliness, good stability, and resistance to aging. Achal et al. [40], Ojha et al. [41], Lv et al. [42], Wang et al. [43], Alvarado-Mata et al. [44], Wang et al. [45], Do et al. [46], Yang et al. [47], Qian et al. [48], Sun [49], and Wei et al. [50] concluded the mineralization mechanism of *S. pasteurii* (Figure 2) and the reaction process is shown in Formulas (1) and (2).
(1)$$CO{\left({NH}_{2}\right)}_{2}+4H_{2}O\;\overset{\mathrm{urease}}{\to }\;2H_{2}O+{CO_{3}}^{2-}+2{NH_{4}}^{+}$$
(2)$$Cell+{Ca}^{2+}+{CO_{3}}^{2-}\to Cell-CaCO_{3}$$

Multiple factors affected the MICP process. Meng et al. [51] and Rui et al. [52] concluded a thorough assessment of these influencing factors in order to optimize MICP technology. Wei [53] and Fan [54] confirmed that *S. pasteurii* had a strong survival ability, being able to grow and reproduce normally at 15–37 ° C, having a pH survival range of 5.5–13, and surviving in the form of spores in the absence of nutrients. Webster et al. showed that the MICP remediation process could eventually form a layer of calcite (CaCO_3_) with a thickness of 5 to 40 μm on the surface [55]. The size of CaCO_3_ crystals can be fully controlled by the rate of urea hydrolysis. The higher the urease activity produced by microorganisms, the faster the rate of urea hydrolysis and the smaller the CaCO_3_ crystals deposited. It is necessary to precisely control the urea hydrolysis rate to obtain the optimal CaCO_3_ crystal morphology and distribution, thereby effectively improving the soil engineering properties.

This study designed a research plan that combined multi-factor experiments with mathematical models. The main objective was to systematically reveal the impact of multiple factors on the characteristics of growth, urease production, and mineralization in MICP processes, establish their importance weights, and derive a comprehensive structural understanding of the mineralization mechanism of *S. pasteurii*.

## 2. Materials and Methods

### 2.1. Study Site

The sampling site of Jinyang Ancient City is located in Jinyuan District, southwest of Taiyuan City, Shanxi Province, China. It includes exposed parts aboveground and buried parts underground. The existing exposed part has a trapezoidal profile with distinct rammed layers, with a total residual length of approximately 800 m. The height is 3–8 m. The width of the bottom is about 15 m, and that of the top is 4–10 m. The vegetation distribution on the city wall was uneven overall. The western section of the wall (Figure 3) supported a mix of shrubs and small trees, while the eastern section was mainly covered with herbs and vines. Those plants had shallow root system, mostly ranging from 200 mm to 300 mm, with weak main roots and dense fibrous roots, forming root–soil composites. Shang et al. [56] and Shang et al. [57] indicated in the soil analysis from the site in Jinyang Ancient City that the predominant soil type was sandy loam, classified as non-collapsible loess. With increasing soil depth, the moisture content showed a corresponding rise. The primary soil salts consisted of CaSO_4_ and MgSO_4_, with pH values ranging from 7.1 to 7.45 (with no unit) and a specific gravity ranging from 2.69 to 2.7 (with no unit given). The volume moisture content of dry soil was about 0.132 m^3^·m^−3^, and the volume moisture content of wet soil was about 0.283 m^3^·m^−3^. The sampling date was 15 May 2022. We did not obtain the permission document, but in 2023 we were in charge of the Shanxi Provincial Cultural Relics Technology Program of China (2023KT15), organized by the Shanxi Provincial Cultural Relics Bureau, which clearly stipulated that the research scope was the Western City wall in Jinyang Ancient City.

### 2.2. Experimental Design

There are numerous factors that affect the growth, reproduction, metabolism and mineralization of *S. pasteurii,* mainly including nutrients, culture conditions, inoculation conditions and mineralization conditions, etc. The main objective of this study was to provide high concentrations of, as well as a high quality of, *S. pasteurii*. In this section, X, U, u, and the yield of calcite in CaCO_3_ were used as different prediction targets. The single-factor comparison experiments and orthogonal experiments were combined to investigate the variation rules of the prediction targets under different conditions, respectively.

There were three common types of *S. pasteurii* culture media, respectively, defined as M_1_, M_2_, and M_3_. The formulation is shown in Table 1. *S. pasteurii* was revived and activated by M_1_, with the culture time sets at 27 h, 54 h, 79 h, and 105 h, respectively. After the successful activation of the above strains, the following experimental work was completed.

The experimental design used for the factors influencing the physiological characteristics of *S. pasteurii* is shown in Table 2. The experiment on the mineralization of CaCO_3_ is shown in Table 3. For the factor experiments, multiple samplings at different time points were conducted according to the experimental plan, and the measurement frequency was also planned. Parameters such as culture temperature, pH value, OD_600_ value, and the difference in the electrical conductivity of *S. pasteurii* bacterial suspension before and after 5 min intervals were recorded. The remaining bacterial suspension in the conical flask during the culture process was kept under the original conditions for continued cultivation. Calcium acetate (CA) was chosen as the calcium source and was provided in solution form. An *S. pasteurii* bacterial suspension with high urease was prepared for standby. According to the experimental requirements, the appropriate liquid mixing method and mineralization method were selected. The precision of the weighing balance was 0.0001 g.

### 2.3. Characterization Parameters

(1)Growth characteristics

The growth curve of *S. pasteurii* is a fluctuating curve plotted with time as the X-axis and *S. pasteurii* bacterial concentration as the Y-axis. It is a fundamental indicator in describing the growth and reproduction rules and is also the core in establishing a growth dynamics model of *S. pasteurii*. The absolute concentration in the model was approximately calculated by OD_600_ with Formula (3) from Wang [23].
(3)$$X=3.283\times {OD}_{600}\times n-0.053$$ where *X* refers to absolute concentration of *S. pasteurii* bacterial suspension, in 10^8^ CFU·mL^−1^; OD_600_ refers to optical density of the bacterial suspension measured at 600 nm, with no unit; and *n* refers to the dilution factor of samples, with no unit.

(2)Urease production characteristics

Urease activity is the ability to hydrolyze urea within a unit of time. Traditionally, the urease activity of *S. pasteurii* was mostly determined qualitatively by colorimetry Xiao (2004) [58], but its quantitative indicators cannot be obtained directly. Urease’s decomposition rate of urea was characterized by measuring the rate of increase in electrical conductivity in the bacterial suspension. The principle was that the increase in CO_3_^2−^ and NH_4_^+^ concentrations during the decomposition of urea led to an increase in the electrical conductivity of the bacterial suspension. The urease activity curve of the *S. pasteurii* was the fluctuation curve of urease activity of the *S. pasteurii* bacterial suspension over time, showing the variation rule of urease activity. The calculation method is shown in Formula (4). Urease Activity Intensity refers to the ability of a unit of *X* to decompose urea within a unit of time. The calculation method is shown in Formula (5).
(4)$$U=f\times \frac{11.11}{0.964}=\frac{\sigma_{1}-\sigma_{2}}{t}\times 1000\times 11.525$$
(5)$$u=\frac{U}{X}$$ where *U* refers to urease activity, in µS·(cm·min)^−1^; f represents the rate of change in electrical conductivity, 2*σ* represents the initial electrical conductivity value, and *σ* represents the electrical conductivity value after *t* minute, in mS·cm^−1^; u refers to the intensity of urease activity, in µS·mL·(CFU·cm·min)^−1^; and *X* refers to the absolute concentration of *S. pasteurii* bacterial suspension, in 10^8^ CFU·mL^−1^.

(3)Mineralization characteristics

The yield of mineralization products was measured by the index parameter method. The actual production of CaCO_3_ was measured on-site. The yield of CaCO_3_ in output was characterized by the ratio of the actual amount of CaCO_3_ to the theoretical production of CaCO_3_, denoted by η, as shown in Formula (6). The formula for calculating the theoretical production of CaCO_3_ is shown in Formula (7). The yield of calcite in CaCO_3_ was characterized by the ratio the actual amount of calcite in CaCO_3_ to the actual amount of CaCO_3_, denoted by η*_cal_*, as shown in Formula (8).
(6)$$\eta =\frac{m_{c}}{m_{0}}\times 100\%=\frac{m_{p}-m_{f}}{m_{0}}\times 100\%$$
(7)$$m_{0}=\min\left(m_{0-a},m_{0-u}\right)=\min\left(\frac{m_{a}}{M_{a}}\times M_{c},\frac{m_{u}}{M_{u}}\times M_{c}\right)=\min\left(0.6328\times m_{a},1.6665\times m_{u}\right)$$
(8)$$\eta_{cal}=\frac{m_{cal}}{m_{c}}\times 100\%=\frac{m_{cal}}{m_{cal}+m_{ara}+m_{vat}}\times 100\%$$ where η is the yield of CaCO_3_ in output; η*_cal_* is the yield of calcite in CaCO_3_, in %; m_c_ is the actual amount of CaCO_3_; *m_p_* is the mass of filter paper with sediment attached; m_f_ is the mass of filter paper; m_0_ is the theoretical amount of CaCO_3_; *m*_0–_*_a_* is the theoretical amount of CaCO_3_ based on the mass of calcium source; *m*_0–_*_u_* is the theoretical amount of CaCO_3_ based on the mass of urea; *m_a_* is the mass of calcium source added before precipitation, *m_u_* is the mass of urea added before precipitation; m_cal_ is the mass of calcite in CaCO_3_; *m_ara_* is the mass of aragonite in CaCO_3_; *m_vat_* is the mass of vaterite in CaCO_3_, in g; *M_a_* is the relative molecular mass of calcium source, 158.17; *M_c_* is the relative molecular mass of CaCO_3_, 100.09; and *M_u_* is the relative molecular mass of urea, 60.06, in g.

### 2.4. Study Methods

The *S. pasteurii* strain exhibited significant individual variability, leading to considerable variation in the results of repeated experiments. Moreover, the influence of each feature on the prediction target for the *S. pasteurii* strain varied greatly, and there was a high likelihood of multi-collinearity among the features. This paper attempted to explore how to use large amounts of data to find the physiological and mineralization characteristics of *S. pasteurii* and provided a reference for future *S. pasteurii* breeding work. A total of 111 items with million-level data were included in the comprehensive statistical experimental group. This feature analysis established a random forest–recursive feature elimination–cross validation model (RF-RFE-CV). The ultimate goal was to obtain the optimal feature set, to master the feature importance weights, and to significantly improve the model’s performance. This paper used the Spyder module in the software of Anacondas 3 in python language with scikit-learn library to implement an optimized multi-objective feature selection analysis model. This model effectively prevented the problem of underfitting by setting a minimum feature selection threshold of at least three important features, using a median imputation strategy to handle missing values, and applying an outlier truncation method to maintain data integrity. It systematically prevented the risk of overfitting through a combination of measures such as limiting the tree number of RF to 100, using ECV to automatically determine the optimal feature subset, implementing 5-fold cross-validation to evaluate the model’s generalization ability, and conducting three repeated permutation importance tests to assess feature stability.

Feature correlation analysis mainly calculated the association strength and direction between each variable. In feature redundancy analysis, the mutual information method for dependency testing was selected to calculate the statistical dependency between features and rearrange the correlation coefficients in the form of a clustering tree. Feature Importance Evaluation (FIE) was based on the two classic methods of impurity reduction importance and permutation importance for research, focusing on the similarities and differences between importance and feature importance. At the same time, we adopted a method of recursive feature elimination, combined with cross-validation, to automatically select the optimal feature subset, improve the generalization ability and interpretability of the model, and enhance the analysis efficiency.

## 3. Results

### 3.1. Feature Importance Evaluation

#### 3.1.1. Longitudinal Analysis

All the data of X, U, and u were taken as research objects to observe the influence patterns of each feature parameter (Table 4). In the importance analysis based on the reduction of impurity, the effects of pH and IA on X and U were significantly overestimated in the situation with no interaction, and the effects of U and pH of X were overestimated in the situation with an interaction. That was because pH, IA, U, etc., had a large number of values, which were typical high-cardinality characteristics. In contrast, the permutation importance method determined the feature importance by measuring the decline degree of model performance after shuffling the feature values, which can effectively prevent the risk of overestimating high cardinality features and intuitively reflect the actual contribution of features to model prediction. Therefore, the subsequent analysis results were based on the permutation importance results.

As shown in Table 4, when the feature parameters of X interacted with U, the number of feature parameter values of X increased from 9 to 10. When the feature parameters of U interacted with X, the number of feature parameters decreased from 8 to 6.

As shown in Figure 4, different parameters had different influences on the prediction results of various prediction targets of *S. pasteurii*. Parameters such as M, B, and G were significantly positively correlated with the prediction results of X. Parameters such as X and B were significantly positively correlated with the prediction results of U. The effects of parameters such as IA, t, pH, and T on the prediction results of X, U, and u were more complex. Specifically, IA and t were positively correlated with the prediction results of the three when there were low and intermediate values, but the positive correlation trend was significantly weakened when there were high values. pH was positively correlated with the prediction results of X and U when taking low and intermediate values, but the negative correlation trend was significantly reduced when taking high values. T had little effect on the three predictions when taking low and intermediate values, but showed significant negative correlation when the temperature was too high.

#### 3.1.2. Cross-Sectional Analysis

On the basis of the above-mentioned feature parameters, the initial pH, initial X, and initial U were added to calculate the feature importance of predicted target feature parameters of *S. pasteurii* at different time points. During the analysis, independent prediction models were constructed for the target variables at each time point. Different feature selection strategies were adopted for various biological indicators. When predicting X, the combined influence of multiple factors and U was considered. When predicting U, the combined influence of multiple factors and X was considered. However, when predicting u, the variables X and U were excluded, and only the influence of multiple factors was considered. This multi-matrix modeling approach could more accurately capture the specific influencing factors of different target variables, avoiding the information redundancy or causal inversion that may result from a single matrix, and enhancing the interpretability and predictive accuracy of the model.

As shown in Figure 5 and Figure 6, and Table 5, the results were as follows. The key feature of X at 4 h was pH-4 h, with a weight of 1.055. The key feature of X at 8 h and 12 h was X-0 h, with weights of 1.477 and 1.317, respectively. The high-importance features of X at 24 h were IV, G, and t, with weights of 0.486, 0.322, and 0.122, respectively. The high-importance features of U at 4 h were X-4 h, pH-4 h, and X-0 h, with weights of 0.589, 0.532, and 0.263, respectively. The key feature of U at 8 h was pH-0 h, with a weight of 1.566. The high-importance features of U at 12 h were IV, pH-0 h, X-12 h, and t, with weights of 0.33, 0.32, 0.248, and 0.125, respectively. The key feature of U at 24 h was X-0 h, with a weight of 1.127. The high-importance features of u at 4 h and 8 h were pH-0 h (with weights of 0.58 for both), pH-4 h (0.316), and pH-8 h (0.282). The high-importance features of u at 12 h were IV and pH-0 h, with weights of 0.882 and 0.112, respectively. The high-importance features of u at 24 h were IV, G, pH-24 h, X-0 h, and pH-0 h, with weights of 0.355, 0.279, 0.28, 0.236, and 0.147, respectively.

### 3.2. Breeding Conditions

#### 3.2.1. Activation Culture Time of Different Media

As shown in Figure 7a–d, the X of *S. pasteurii* in M_1_, M_2_ and M_3_ reached its maximum at 105 h, U in M_1_ and M_3_ reached its maximum with activation culture time at 27 h, and U in M_2_ reached its maximum with activation culture time at 54 h.

#### 3.2.2. Temperature

As shown in Figure 7e,f, when using M_1_, the temperature at which X reached its maximum within 24 h was 30 °C, and the temperature at which urease reached its maximum activity within 24 h was 32 °C. When using M_2_, the maximum X was around 37 °C and the maximum activity of urease was around 30 °C within 24 h.

#### 3.2.3. pH Value

The pH of the *S. pasteurii* bacterial suspension changed over time according to the following patterns. The pH value within the range of 5.5–10 was within the overall tolerance range of *S. pasteurii* bacterial suspension (Figure 8). The λ values of the *S. pasteurii* bacterial suspension at pH 5.5–7 and 10 were significantly longer than those at pH 7.25–9. That was, adjusting the pH to 7.25–9 could significantly shorten the lag period of *S. pasteurii*. As shown in Figure 7g–i, under M_1_ conditions, X was maximized at around pH 7.25, and U was maximized at pH 8. Under M_2_ conditions, X was at its maximum at pH 7, and U was at its maximum at pH 9. Under M_3_ conditions, X was at its maximum at pH 7, and U was at its maximum at pH 8.

#### 3.2.4. Dissolved Oxygen

As shown in Figure 7j, peak X and U values both occurred at 170 r/rpm. Therefore, to obtain higher U, it was recommended to select an oscillation speed of 170 r/rpm for M_1_.

#### 3.2.5. Inoculation Age

Different IAs lead to inconsistent X. When we used the same volume with different X of *S. pasteurii*, it inevitably caused inconsistent IV. But if OD_600_ values were adjusted, the effect of T on IA was diluted. Therefore, there was a certain collinearity between IA and IV.

As shown in Figure 7k, X of M_1_ reached its maximum at 48 h of IA, and U reached its maximum at 16 h of IA.

#### 3.2.6. Inoculation Volume

As shown in Figure 7l, it can be seen that when liquid medium was used as *S. pasteurii* source, X of M_1_ reached the maximum at 3% and U reached the maximum at 1%. Therefore, in order to obtain higher U, it was recommended to choose IV of 1% for M_1_.

To sum up, the optimal culture conditions for X, U and u are shown in Table 6.

### 3.3. Mineralization Conditions

When the yellow *S. pasteurii* bacterial suspension was mixed with the clear and transparent calcium source, large quantities of white turbid substances rapidly appeared in the mixture. As shown in Figure 9a–d, within 30 min, the mass of the mineralization products generated by the calcium acetate solution increased rapidly. After 30 min, the growth rate of its mass gradually slowed down with the extension of time. It can be seen that the mineralization of *S. pasteurii* was not completed instantaneously but was a continuous process.

As shown in Figure 10a, the mineralization products were filtered, dried, and analyzed by EDS spectrum. It was found that the three elements of mineralization products were Ca:C:O ≈ 1:1:3, which fully confirmed that the mineralization product of *S. pasteurii* was calcium carbonate (CaCO_3_).

#### 3.3.1. Ca^2+^ Concentration

(1)Experiment with a molar ratio of 1:1

Figure 11 showed the changes in mineralization products over time at different Ca^2+^ concentrations. In the initial stage, the precipitation rate of mineralization products of *S. pasteurii* was relatively high. As time increased, the mineralization products accumulated, but their precipitation rate slowed down significantly.

The Ca^2+^ concentration was positively correlated with the net mass of CaCO_3_ and negatively correlated with the yield of CaCO_3_ in output. Within the same period of time, the higher the Ca^2+^ concentration, the greater the net mass of the mineralization products of CaCO_3_. The yield of CaCO_3_ in terms of output followed the pattern η_0.7_ > η_0.9_ > η_1_ > η_1.1_ > η_1.3_.

If rapid restoration is needed, to obtain the same amount of CaCO_3_, the Ca^2+^ concentration can be increased, which can greatly reduce the time required for mineralization, but the yield of CaCO_3_ in the output will be significantly reduced. In the absence of specific requirements for restoration time, a low Ca^2+^ concentration was significantly more effective than a high Ca^2+^ concentration in achieving higher CaCO_3_ yield in the output.

(2)Experiment with a molar ratio of n:1

As shown in Figure 12, it can be seen that the higher the Ca^2+^ concentration was, the greater the U of *S. pasteurii* was. This indicated that the high Ca^2+^ concentration could significantly stimulate the U of *S. pasteurii*. As the Ca^2+^ concentration increased, the net mass of CaCO_3_ gradually increased and reached a peak at 1.3 M, and then gradually decreased. When the Ca^2+^ concentration was less than 1 M, the yield of CaCO_3_ in output decreased gradually with the increase in Ca^2+^ concentration. When the Ca^2+^ concentration was no less than 1 M, the yield of CaCO_3_ in output increased first and then decreased gradually with the increase in Ca^2+^ concentration. The reason for the above phenomenon was speculated to be related to the 1 M urea concentration. The above regularity conformed to the short-board effect, and the low value of the theoretical value determined its fluctuation pattern.

#### 3.3.2. Addition Order and Reaction Condition

As shown in Figure 9e–l, the comparison experiments of the on-site observation of *S. pasteurii* addition revealed that all mixed solutions became white at the beginning, while the mixed solutions of Group B and Group D had higher transparency than Group A and Group C at 24 h. At 48 h, it was observed that the precipitates of Group A and Group C gathered at the bottom of the bottle, while those of Group B and Group D were attached to both the bottle wall and the bottom.

As shown in Figure 13, by weighing the net mass of the products, the following results can be obtained. Overall, the net mass of CaCO_3_ obtained from the four experimental groups followed the order: Group A > Group C > Group D > Group B. It can be seen that the net mass of CaCO_3_ in the premix method of *S. pasteurii* was significantly higher than that of the post-add method of *S. pasteurii*. In the same mixture, the culture method had little effect on the net quality of the mineralization product. Therefore, in order to increase the mineralization yield, it was recommended to use the premix method of *S. pasteurii*.

The SEM images of the mineralization products of Group B, Group C, and Group D are shown in Figure 10b–d. The XRD patterns are shown in Figure 10e. Both the mineralization products of Group B and Group D were mixed crystals of vaterite and calcite, while the mineralization product of Group C was calcite crystals. Quantitative analysis of the samples was conducted using Jade 9 software (Table 7). It was found that after 96 h of reaction, no aragonite was produced in the mineralization products. By comparing samples of Group B and Group D, it was observed that the two different mixing methods, quiescent and shaking, had little effect on the yield of calcite in CaCO_3_. Both produced a mixture of vaterite and calcite, and the proportion of vaterite was much higher than that of calcite. By comparing the data of Group C and Group D, the yield of calcite in CaCO_3_ of premix method reached 100%, while the one of post-add method was only 5.6%.

#### 3.3.3. Filtration and Drying Methods

(1)Quality of mineralization products

The quality of calcium acetate solution under different filtration and drying methods (Table 8) was as follows. For a 0.5 M calcium acetate solution, both the net mass and the yield of CaCO_3_ in the output followed the order: Group III > Group II > Group IV > Group I. For a 0.9 M solution, the same parameters followed the order: Group I > Group IV > Group III > Group II. For a 1.0 M solution, the order was Group III > Group II > Group I > Group IV. For a 1.1 M solution, the order was Group III > Group IV > Group I > Group II. For a 1.5 M solution, the order was Group I > Group II > Group IV > Group III. Across all concentrations, the yield of CaCO_3_ in the output ranged from 60.4% to 68.7%. Overall, the net mass of CaCO_3_ followed the order: Group I > Group IV > Group II > Group III.

(2)Crystal form of mineralization products

The XRD patterns are shown in Figure 10e. The mineralization products of Group I, Group II, Group III, and Group IV were all mixed crystals of vaterite and calcite. Through quantitative analysis of the samples by Jade software (Table 7), it was discovered that the yield of vaterite in CaCO_3_ was significantly higher than that of calcite in CaCO_3_ in the samples from the above four experimental groups.

## 4. Discussion

Although this study was based on only 111 sets of experimental conditions, the total amount of raw data reached the million-level, meeting the basic requirements of big data analysis. The core contribution of this paper lies in proposing and validating a random forest methodological framework to provide references enabling subsequent research to achieve more accurate and interpretable analysis and prediction of mineralization mechanisms.

### 4.1. S. pasteurii Features of Urease Production

(1)Longitudinal Analysis

When the feature parameters of X interacted with U, the values of the feature parameters of X increased from 9 to 10, and the explanatory power of each feature parameter changed. The reason for this phenomenon was that there was a strong correlation between U and some of the feature parameters of X, which led to the dispersion of its importance. However, too many interaction parameters can significantly reduce the explanatory power among parameters. Therefore, it was not recommended to consider U as an important feature parameter of X.

When the feature parameters of U interacted with X, the number of feature parameters decreased from 8 to 6, and the explanatory power of each feature parameter changed. As shown in Figure 7, the peak of U of *S. pasteurii* occurred earlier than X. It can be seen that the stress response of U to harmful metabolites was earlier than that of X. The reason for this phenomenon was that X was measured by OD_600_ value, which was based on the principle of light scattering and absorption by the suspension. However, unlysed dead bacteria retained the integrity of the cell structure and still scattered and absorbed light, making it impossible to distinguish between live and dead cells, resulting in the change trend of U of *S. pasteurii*. It can be seen that urease was produced by *S. pasteurii* to meet its own growth requirements, and when the content of external urea increased, it stimulated *S. pasteurii* to produce more urease, thereby increasing the mineralization yield. In other words, the growth of the *S. pasteurii* microbiota was the foundation, and the increase in urease was a downstream product of environmental stimulation.

In summary, when considering all the feature parameter data of *S. pasteurii* throughout its longitudinal analysis, it was not recommended to interact with U for each feature parameter of X. Among the feature parameters of X, the critical feature was t (1.517), and the high-importance features followed the order of IV (0.255) > M (0.21) > IA (0.208) > pH (0.173). Among the feature parameters of U, the key feature was X (1.016), and the ranking of the high-importance features followed the order of t (0.236) > IA (0.213) > B (0.177). The ranking of key features for u was t (1.341) > IA (1.234), and the high-importance feature was pH value (0.916).

(2)Cross-sectional analysis

Cross-sectional analysis revealed that at 4 h, there was a significant correlation between X and pH-4 h, with a weight of 1.055. At 8 h and 12 h, the impact of X-0 h emerged as a crucial feature, with weights of 1.477 and 1.317, respectively. It was consistent with the growth phase of *S. pasteurii*, indicating that X followed a logarithmic growth pattern during the rapid-growth stage. In the slow-growth stage, at 24 h, multiple factors such as IV and G governed the changes in X, with weights of 0.486 and 0.322, respectively. This phenomenon can be ascribed to the exhaustion of nutritional substrates and the accumulation of metabolic by-products in the late stage of cultivation, which led to a deceleration of the growth trend. The findings of this study were highly congruent with the typical growth curve of *S. pasteurii* under limited nutritional conditions and were mutually corroborated by the research results of Wang [23] and Wei [53], thereby enhancing the reliability and general applicability of the conclusion.

At 4 h, U did not exhibit a single dominant characteristic. The weights of the main influencing factors were arranged in the following order: X-4 h (0.589) > pH-4 h (0.532) > X-0 h (0.263). At 8 h, pH-0 h became a crucial characteristic for regulating U, and its weight increased significantly to 1.566. The most important feature was X-8 h, with a weight of 0.242. It was well known that urease, as an enzyme associated with bacterial metabolism, had an activity closely related to X. A higher X-8 h can enhance the overall U. At 12 h, U was once again influenced by multiple features, and no single key feature was detected. At this time, IV, pH-0 h, X-12 h, and t were identified as features of high importance, with weights of 0.330, 0.320, 0.248, and 0.125. After 24 h of cultivation, X-0 h once again became the decisive feature influencing U, with a weight of 1.127, indicating that the dominant role of this parameter on U was reinforced during the slow-growth period.

The u of *S. pasteurii* was primarily regulated by pH at 4 h and 8 h. The weight of the initial pH value was 0.580 at both time points, with weights of 0.316 and 0.282, respectively. This result fully indicated that pH played a decisive role in u during the lag phase and the exponential growth phase of *S. pasteurii*. At 12 h, the dominant feature of u underwent a shift. IV became the most critical feature, with a weight of 0.882, while the influence of pH-0 h value weakened significantly, with a weight of 0.112. This indicated that the inoculation scale of *S. pasteurii* replaced pH as the main regulatory factor. After 24 h of cultivation, u presented a pattern of being co-regulated by multiple features, and the weight distribution of each parameter tended to be balanced. IV, G, pH-24 h, X-24 h, and pH-0 h formed a set of features of high importance, with weights of 0.355, 0.279, 0.280, 0.236, and 0.147, respectively, reflecting the complexity of the u regulation mechanism of *S. pasteurii*.

Evidently, the selection and breeding of *S. pasteurii* was a dynamic process. The importance and the positive or negative effects exerted by the same feature parameter may vary significantly, or even be opposite, across different prediction targets and cultivation time points. This refuted the simplistic assumption that a particular characteristic was always the most important or always had a promoting effect. It confirmed that the importance and influence direction of characteristics for different prediction targets exhibited a high degree of spatiotemporal specificity and goal dependence. Therefore, the optimization strategy must involve precise regulation tailored to specific time points and goals, rather than a fixed formula.

(3)Comprehensive analysis

It was known that U was a core functional indicator for predicting and evaluating the mineralization efficiency of *S. pasteurii*. Longitudinal analysis (Table 4) showed that when the characteristic parameters of U interacted with X, the number of characteristic parameters reduced from 8 to 6, and X (1.016) became a key characteristic of *S. pasteurii*, which was far more important than other characteristic parameters. Ignoring the continuity of t, the cross-sectional data (Table 5) of *S. pasteurii* showed that the weights of X-4 h and X-0 h at 4 h were 0.589 and 0.263, respectively; the weight of X-8 h at 8 h was 0.242; the weight of X-12 h at 12 h was 0.248; and the weights of X-0 h and X-24 h at 24 h were 1.127 and 0.259, respectively. X was a high-importance feature at multiple time points such as at 4 h, 8 h, 12 h and 24 h, and even at 24 h, X-0 h was a key feature. This meant that the best response time was at around 24 h, which proved that urease activity was highly dependent on the initial X in the later stage of culture, and the growth potential in the early stage was fully reflected at this stage. In other words, X was a key driving factor for characterizing U, and its inclusion in the U prediction model could reduce the number of characteristic parameters and improve the explanatory power of each characteristic parameter. Its importance was confirmed in both longitudinal and cross-sectional data analysis. It can be seen that U was closely related to the initial pH value and the number of bacteria. A higher initial X could prompt *S. pasteurii* to enter the rapid proliferation period faster, which would promote an overall increase in X and U. The above conclusion was in line with the general law of bacterial growth.

Longitudinal analysis (Table 4) indicated that t (1.517) was the most critical factor influencing X, with the ranking of highly influential factors following the rule of IV (0.255) > M (0.21) > IA (0.208) > pH (0.173). Cross-sectional analysis revealed distinct temporal patterns in feature importance. pH-4 h and pH-0 h at 4 h exhibited weights of 1.055 and 0.189, respectively, while X-0 h had a weight of 0.154. pH-0 h value at 8 h decreased to 0.498, whereas X-0 h increased significantly to 1.477. pH-0 h and pH-12 h at 12 h were weighted at 0.42 and 0.116, respectively, with X-0 h remaining high at 1.317. The influence of pH and X-0 h at 24 h diminished, while IA and G emerged as dominant factors, with weights of 0.486 and 0.322, respectively. Redundancy analysis of bacterial concentration-related features demonstrated a redundancy weight of 1.052 between t and pH and one of 0.917 between IA and IV, suggesting potential multi-collinearity issues. In the growth curve of *S. pasteurii*, pH was identified as the key determinant during the lag phase, with pH-4 h and pH-0 h at 4 h weighted at 1.055 and 0.189, respectively, indicating strong regulation by the initial culture medium environment. The results of pH experiment demonstrated that maintaining pH within the range of 7.25–9 could effectively shorten the duration of the lag phase, further validating the decisive influence of pH on X during the lag phase. During the exponential growth phase, X-0 h at 8 h and 12 h became the predominant factor, with weights of 1.477 and 1.317, aligning well with the expected growth kinetics of *S. pasteurii* during logarithmic proliferation. This highlighted the decisive impact of inoculation strategy on U synthesis in this phase. In contrast, during the stationary or deceleration phase (24 h), influencing factors became more diverse, with weights of 0.486 and 0.322 on IV and G increasing in relative importance, reflecting a complex regulatory stage of bacterial growth. This shift was likely attributable to nutrient depletion and the accumulation of metabolic by-products in later culture stages, which collectively suppress growth rates. These findings were in close agreement with the established growth profile of *S. pasteurii* under nutrient-limited conditions and corroborate previous reports by Wang [23] and Wei [53], thereby enhancing the reliability and generalizability of the conclusions.

In addition to X, the ranking of high-importance features for U in the longitudinal analysis followed the order of t (0.236) > IA (0.213) > B (0.177). In the cross-sectional analysis, U exhibited pH-4 h at 4 h showing a weight of 0.532. pH-0 h at 8 h reached the highest observed weight of 1.566. pH-0 h at 12 h registered at 0.32 and pH-24 h at 24 h contributed a weight of 0.109. These results indicated that pH was a key determinant of *S. pasteurii* in U, particularly during the early exponential growth phase. The peak importance of pH-0 h at 8 h underscored its critical role in modulating U, further validating pH as a major influencing factor in cross-sectional analysis.

The u served as a core quality indicator for assessing U, being widely utilized in the screening of high-efficiency strains and the optimization of culture conditions. Longitudinal analysis revealed that the key determinants of u were ranked in descending order of culture time (1.341) > inoculation age (1.234), with pH emerging as a highly influential factor (0.916). Cross-sectional analysis demonstrated distinct temporal patterns in regulatory influences. During the early growth phase, u in *S. pasteurii* was predominantly governed by pH conditions. pH-0 h exhibited a consistent weight of 0.580 at 4 h and 8 h, while pH-4 h and pH-8 h registered weights of 0.316 and 0.282, respectively. As cultivation progressed at 12 h and 24 h, the influence of IV increased markedly, with importance weights of 0.882 and 0.355. Concurrently, the role of pH diminished—pH-0 h at 12 h dropped to 0.112, and pH-24 h and pH-0 h at 24 h contributed only 0.280 and 0.147, respectively. Redundancy analysis of u revealed a redundancy weight of 0.956 between IA and IV, suggesting a significant degree of multi-collinearity. Cross-sectional evaluation identified pH and IV as primary high-importance features. The significance of pH was consistently validated across multiple time points, whereas the impact of IV became prominent after 12 h, highlighting its growing regulatory role in the mid-to-late culture phase.

### 4.2. S. pasteurii Features of Mineralization

(1)The yield of CaCO_3_ in output

The research on Ca^2+^ concentration confirmed that the yield of CaCO_3_ in output was higher when the same low concentrations of calcium source and urea were used. Ghosh et al. [59] found that an excessively high enzymatic hydrolysis rate was not conducive to enhanced mineralization, and a slow and stable rate of calcite mineralization deposition was more beneficial for the enhancement effect of carbonate mineralization bacteria. Ahmad et al. [60] and Xu et al. [61] indicated that this broad adaptability could not be directly translated into the optimal efficiency of the MICP process. Blindly pursuing high urease activity might be counterproductive. The net mass and the yield of CaCO_3_ in the output were higher at a Ca^2+^ concentration of 1.3 M compared to 1.0 M. It can be seen that when the urease activity of *S. pasteurii* was constant, the ratio of urea concentration to Ca^2+^ concentration at 1 M:1.3 M was more conducive to the formation of high-quality net mass and the yield of CaCO_3_ in output.

As shown in Table 8, with the filtration methods, centrifugal filtration was faster than undisturbed filtration, but the yield of CaCO_3_ in output followed the pattern of undisturbed filtration > centrifugal filtration, with an error of 5.2% to 5.4%. With the drying method, oven drying was quicker than room-temperature drying in the shade, but the yield of CaCO_3_ in output followed room-temperature drying in the shade > oven drying, with an error of 3% to 3.1%. However, based on the fact that centrifugal filtration could quickly and thoroughly collect all solid precipitates onto the filter paper, in many cases, considering the time-saving aspect, centrifugal filtration and oven drying were often adopted to speed up the process, with an error of about 8.3%.

Therefore, to maximize the yield of CaCO_3_ in output, it was recommended to follow the condition of the ratio of urea concentration to Ca^2+^ concentration at 1 M:1.3 M, undisturbed filtration, and room-temperature drying in the shade.

(2)The yield of calcite in CaCO_3_

As shown in Figure 10b–d, the crystals of the mineralization products in group B were mostly needle-like, while those in group D were stacked to form disk-shaped structures ranging from 10 to 20 μm. Group C presented granular and blocky forms, with some crystals bonded together, and the crystal size ranged from 2 to 15 μm. This indicated that the addition order of *S. pasteurii* had an impact on the appearance of the mineralization products. By comparing the data of Group B, Group C and Group D in Table 7 and the samples in Figure 10e, it was determined that different addition methods of *S. pasteurii* had different impacts on the crystal form of the mineralization product. The premix method was most conducive to improving the yield of calcite in CaCO_3_, which could reach 100%.

As shown in the relevant data of Group I, Group II, Group III, and Group IV in Table 7, centrifugal filtration and room-temperature drying in the shade was more likely to produce calcite. Among the filtration methods, the yield of calcite in CaCO_3_ followed the order of centrifugal filtration > undisturbed filtration, and the error range was from 0.9% to 1.1%. Among the drying methods, the yield of calcite in CaCO_3_ followed the order of room-temperature drying in the shade > oven drying, and the error range was from 1.8% to 2%.

Therefore, to maximize the yield of calcite in CaCO_3_, it is recommended to use the following conditions: using the premix method of *S. pasteurii*, quiescent reaction, centrifugal filtration, and drying at room-temperature in the shade environment.

(3)Comprehensive analysis

A comparison of the recommended conditions for the yield of CaCO_3_ in output and the yield of calcite in CaCO_3_ revealed that the conflicting condition was the filtration method. The yield of CaCO_3_ in output needed undisturbed filtration, while the yield of calcite in CaCO_3_ needed centrifugal filtration.

The lower yield of CaCO_3_ in output observed in centrifugal filtration may be attributed to the loss of tiny particles during the post-centrifugation process. The high gravity during centrifugation effectively precipitated larger and well-crystallized particles. However, it also caused some tiny CaCO_3_ precursors to be lost along with the supernatant when poured onto the filter paper. The higher yield of calcite in CaCO_3_ observed in centrifugal filtration may be attributed to the fact that calcite particles were larger in size compared to the particles of aragonite and vaterite, making them more likely to remain on the filter paper.

Among the mineralization products obtained from Group I, II, III and IV, the yields of calcite in output were 4.27%, 4.98%, 4.15%, and 2.65%, respectively. Under the same condition of room-temperature drying in the shade, the difference in calcite content between undisturbed filtration in Group I and centrifugal filtration in Group II was only 0.71%, indicating no significant difference in crystal form selectivity between them. Meanwhile, the yield of CaCO_3_ in output with the condition of undisturbed filtration was higher than that of centrifugal filtration by 5.2–5.4%, demonstrating a clear advantage in output. Therefore, considering the stability of output, the condition of undisturbed filtration was more appropriate.

In summary, to maximize the yield of CaCO_3_ in output and the yield of calcite in CaCO_3_, we recommend the following conditions: the ratio of urea concentration to Ca^2+^ concentration of 1 M:1.3 M, using the premix method of *S. pasteurii*, quiescent reaction, undisturbed filtration, and drying at room-temperature in the shade environment.

## 5. Conclusions

To understand the cultivation and characteristics of *S. pasteurii*, this paper conducted multiple experiments on *S. pasteurii*. The random forest model method was used to analyze the feature importance weight of data. Optical microscopy, scanning electron microscopy, X-ray computed tomography, and image analysis were used to observe the mineralization process. The feature parameters of the growth, urease production, and mineralization of *S. pasteurii* were optimized. The culture scheme of *S. pasteurii* with high urease activity was obtained, and mineralization conditions with high yield of CaCO_3_ in output and high yield of calcite in CaCO_3_ were obtained. The conclusions were as follows.

(1)Parameter optimization.

The U of *S. pasteurii* was primarily regulated by three key parameters with X, pH, and u, which required stage-specific dynamic control throughout the growth cycle. During the optimization of U, X consistently emerged as a high-importance feature across multiple time points, with peak effectiveness observed at 24 h (1.127). However, the regulatory factors of X varied across growth phases. Across the lag, rapid-growth, and slow-growth phases, the dominant influence on X progressively shifted from pH to X-0 h, and ultimately to a broader set of regulatory factors, reflecting increasingly complex physiological and environmental interactions. pH remained a highly influential parameter across several time points, exhibiting maximum impact at around 8 h (1.566). With u, the significant factors were pH and IV. pH exerted a pronounced influence during the early cultivation stage, whereas IV gained increasing importance after 12 h, underscoring its role in sustaining metabolic activity in mid-to-late culture phases. The above conclusion would provide important experimental data references in actual soil where temperature and pH values fluctuate for scaling up or field formulation.

(2)Scheme for improving urease activity.

To obtain higher urease activity, it was recommended to prioritize the use of Medium 220. CASO AGAR (Merck 105458) as the culture medium for *S. pasteurii*. The activation culture time was 27 h. The inoculation age was 16 h. The inoculation amount was 1%. The culture temperature was set at 32 °C. The initial pH value was 8. And oscillation speed was 170 r/min.

(3)Scheme for enhancing the yield of mineralization.

To maximize the yield of CaCO_3_ in output and the yield of calcite in CaCO_3_, it was recommended to follow the condition of the ratio of urea concentration to Ca^2+^ concentration at 1 M:1.3 M, premix method of *S. pasteurii*, quiescent reaction, undisturbed filtration, and room-temperature drying in the shade.

This study was based on systematic experimental data and introduced the random forest algorithm for big data analysis. It not only achieved a comparative analysis of the influence of single factors but also revealed the importance weights of multiple factors on MICP from a macro perspective. The aim of this data analysis method was to help readers develop a comprehensive and structural understanding of the mineralization influence mechanism, and to provide precise guidance for the restoration technology for splitting fissures of root–soil composites covered with primary vegetation.

## Figures and Tables

**Figure 1 microorganisms-13-02802-f001:**
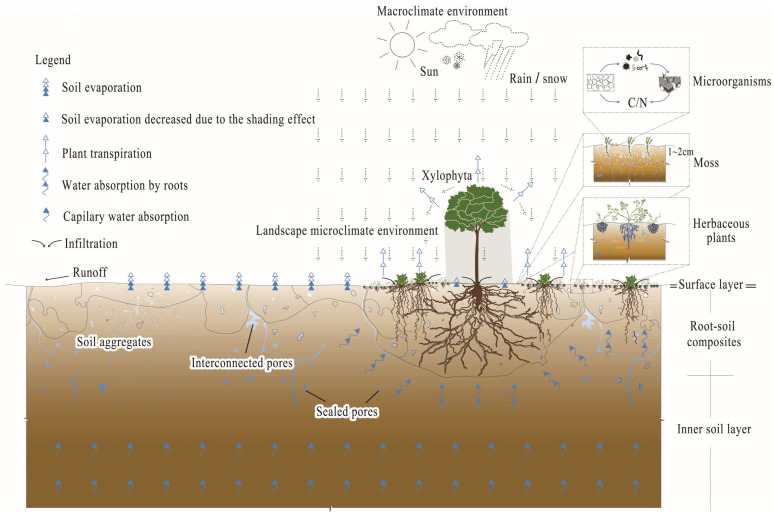
Schematic diagram of wet migration with earthen sites and plants.

**Figure 2 microorganisms-13-02802-f002:**
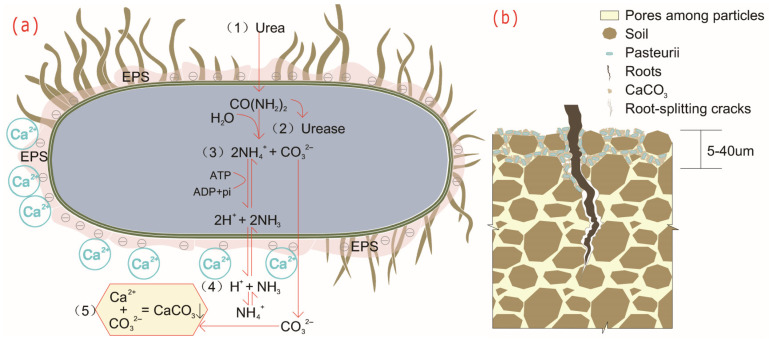
Schematic diagram of *S. pasteurii* mineralization deposition and crack restoration. Mineralization deposition process (**a**); crack restoration (**b**).

**Figure 3 microorganisms-13-02802-f003:**
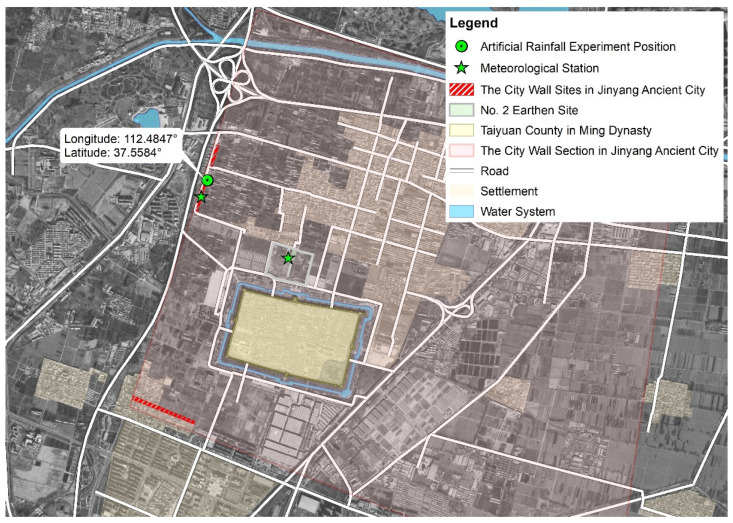
Geographical location of study areas in Jinyang Ancient City.

**Figure 4 microorganisms-13-02802-f004:**
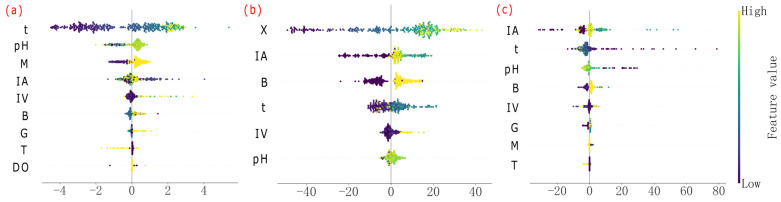
Distribution of SHAP values for feature parameters of each predicted target. X (**a**); U (**b**); u (**c**).

**Figure 5 microorganisms-13-02802-f005:**
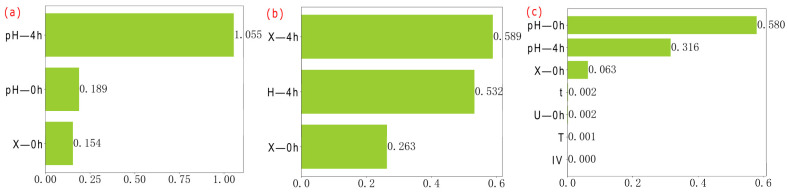
Distribution of feature importance for each prediction target at different time points. X-4 h (**a**), U-4 h (**b**), u-4 h (**c**), X-8 h (**d**), U-8 h (**e**), u-8 h (**f**), X-12 h (**g**), U-12 h (**h**), u-12 h (**i**), X-24 h (**j**), U-24 h (**k**), u-24 h (**l**).

**Figure 6 microorganisms-13-02802-f006:**
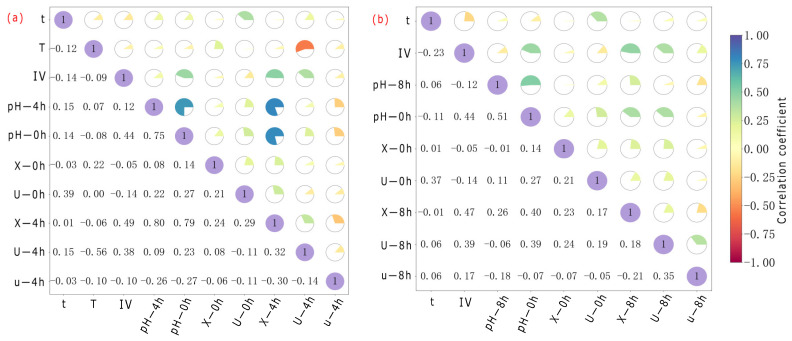
Heat map of the correlation of feature parameters of *S. pasteurii* at different time points: 4 h (**a**), 8 h (**b**), 12 h (**c**), 24 h (**d**). The purple circle with number 1 represents the correlation of a variable with itself. It indicates a perfect positive correlation between them.

**Figure 7 microorganisms-13-02802-f007:**
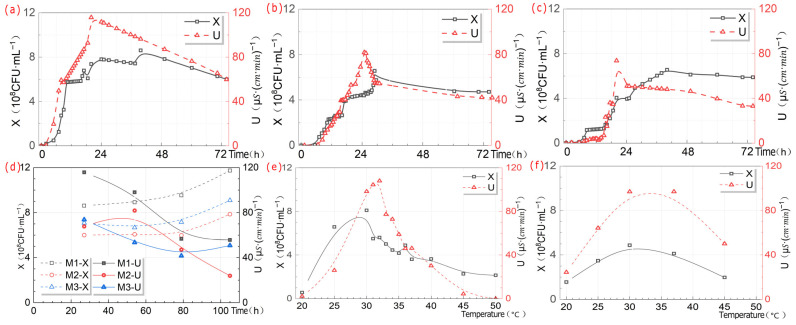
Trend charts of the effects of each feature parameter on each prediction target of *S. pasteurii*. M_1_—X and U with activation culture time at 27 h (**a**); M_2_—X and U with activation culture time at 54 h (**b**); M_3_—X and U with activation culture time at 27 h (**c**); M_1_, M_2_ and M_3_—M at different ACt (**d**); M_1_—X and U at different T (**e**); M_2_—X and U at different T (**f**); M_1_—X and U at different pH (**g**); M_2_—X and U at different pH (**h**); M_3_—X and U at different pH (**i**); M_1_—X and U at different DO (**j**); M_1_—X and U at different IA (**k**); M_1_—X and U at different IV (**l**).

**Figure 8 microorganisms-13-02802-f008:**
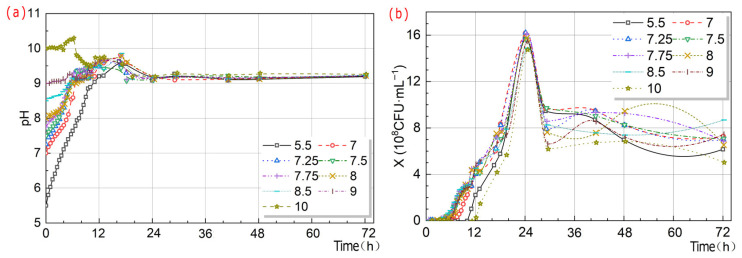
Schematic diagram of pH value and X over time by *S. pasteurii*. pH value over time (**a**); X over time (**b**).

**Figure 9 microorganisms-13-02802-f009:**
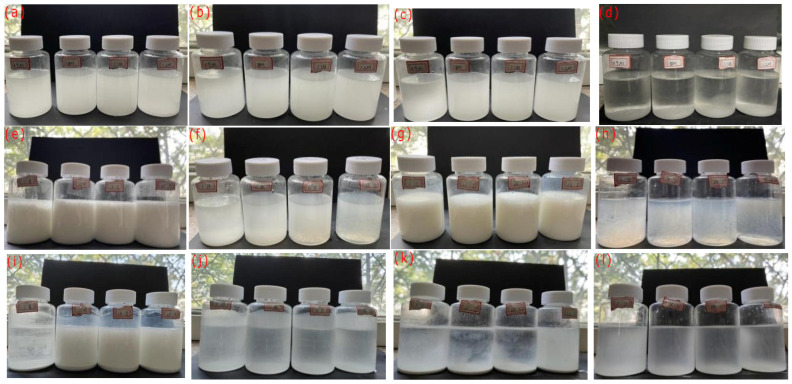
Mineralization process by *S. pasteurii*: 0 min (**a**), 10 min (**b**), 30 min (**c**), 12 h (**d**), Group A—24 h (**e**), Group B—24 h (**f**), Group C—24 h (**g**), Group D—24 h (**h**), Group A—48 h (**i**), Group B—48 h (**j**), Group C—48 h (**k**), Group D—48 h (**l**).

**Figure 10 microorganisms-13-02802-f010:**
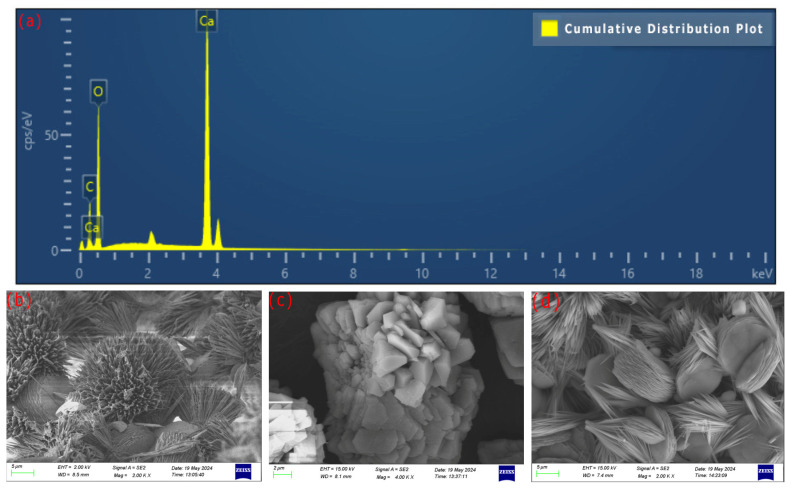
Spectrum diagram of mineralization product of *S. pasteurii*. EDS (**a**), Group B—SEM (**b**), Group C—SEM (**c**), Group D—SEM (**d**), XRD (**e**).

**Figure 11 microorganisms-13-02802-f011:**
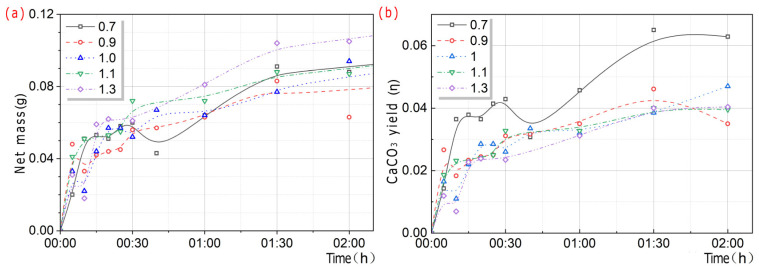
Trend charts in mineralization products over time at different Ca^2+^ concentrations. Net mass (**a**), CaCO_3_ yield (**b**).

**Figure 12 microorganisms-13-02802-f012:**
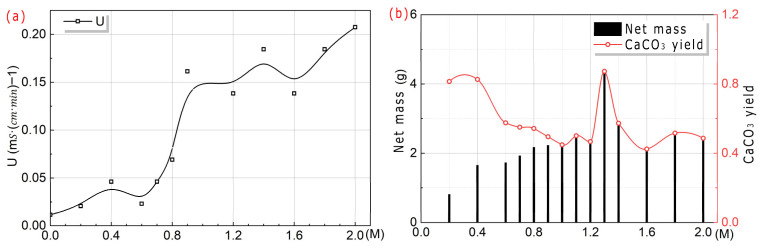
Trend charts with U and CaCO_3_ at different Ca^2+^ concentrations. Ca^2+^ concentration and U (**a**), net mass and yield of CaCO_3_ in output (**b**).

**Figure 13 microorganisms-13-02802-f013:**
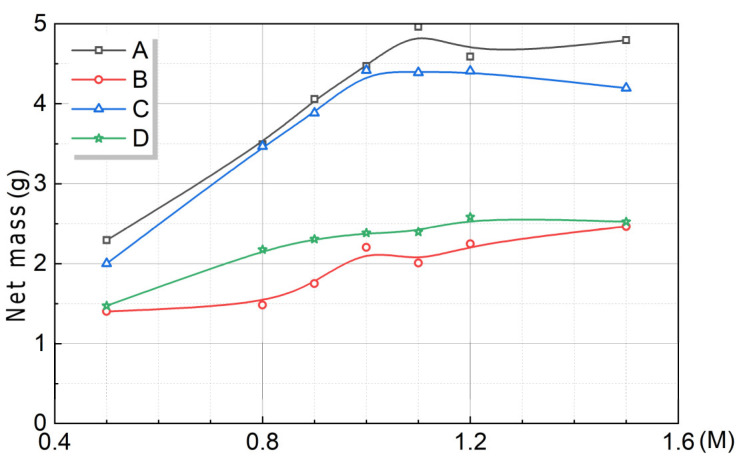
Trend chart in net mass with Ca^2+^ concentration under different addition orders.

**Table 1 microorganisms-13-02802-t001:** Formulation of various culture media used in the experiments.

Names		M_1_		M_2_		M_3_	
		CASO AGAR Medium 220		Medium LB		ATCC Medium 1376	
Nitrogen source	Organic	Peptone from casein	15 g	Peptone	5 g	Yeast extract	20 g
		Peptone from soymeal	5 g	Beef extract	3 g		
		Urea *	20 g	Urea	20 g	Urea	20 g
	Inorganic	— **	—	—	—	(NH_4_)_2_SO_4_	10 g

Inorganic salts		NaCl	5 g	—	—		

						NaCl	1 g

Solvent		Deionized water	1000 mL	Deionized water	1000 mL	Deionized water	1000 mL
pH value		pH	7.3	pH	7.0	pH	7.3
Buffer system		—	—	—	—	Tris buffer	0.13 mol
Gelling agent ***		Agar	15 g	Agar	15 g	Agar	15 g

Note: * There is no urea in the official medium. Wei [53] confirmed that the addition order of urea concentration to the medium could significantly stimulate *S. pasteurii* to produce more urease, thereby increasing the mineralization yield by *S. pasteurii*. ** “—” indicates that the factor is empty. *** The gelling agent is only used in a solid medium and does not need to be added to a liquid medium.

**Table 2 microorganisms-13-02802-t002:** Experimental scheme for factors affecting the physiological characteristics of *S. pasteurii*.

Experiment Names		Parallel Experiments	Replicates	Culture Capacity (mL)		G	T (°C)	Initial pH		DO (r/min)		IA (h)		IV (%)		t (h)	
ACt	M_1_	1	4	300		1	30	7.3		170		/		10~100 µg		72	
	M_2_	1	4					7									
	M_3_	1	4					7.3									
T	M_1_	13	2	100		2	/	7.3		170		24		2		24	
	M_2_	6	2					7									
pH	M_1_	14	1	100		1	30	/		170		27		2		72	
	M_2_	3	1													24	
	M_3_	9	1														
DO	M_1_	7	1	100		2	30	7.3		/		24		2		24	
IA	M_1_	16	1	200			30	7.3		170		/		1		48	
IV	M_1_	16	1	100		3	30	7.3		170		24		/		24	
**Experiment Names**		**Experimental conditions**															
ACt (h)	M_1_	27			54			79			105			—			
	M_2_																
	M_3_																
T (°C)	M_1_	20	25	30	31	32	33	34	35	36	37	40	45	50	—		
	M_2_	20	25	30	—	—	—	—	—	—	37	40	45	—			
pH	M_1_	—	5.5	6	7	7.25	7.3	7.5	7.75	8	8.5	9	10	11	12	13	—
	M_2_	—	—	—	7	—	—	—	—	8	—	9	—	—	—	—	
	M_3_	5	—	6	7	—	—	—	—	8	—	9	10	11	12	13	
DO (r/min)	M_1_	0		50		100		150		170		200		250		—	
IA (h)	M_1_	11	12	13	14	15	16	17	18	19	20	22	24	25	27	28	30
IV (%)	M_1_	9	10	11	12	13	14	15	16	0.5	1	1.5	2	2.5	3	3.5	4

Experiment name explanation: ACt refers to the activation culture time of different media. T refers to the culture temperature of media. pH refers to the pH value. DO refers to dissolved oxygen. IA refers to the inoculation age of *S. pasteurii*. IV refers to the inoculation volume of *S. pasteurii*. t refers to the cultivation time of mixture. G refers to the cultivation generation of *S.*
*pasteurii*. Symbol explanation: “/” indicates a single-factor element, and “—” indicates that the factor is not measured. Experimental group description: The optimization design involved a total of 115 experiments, among which 111 sets of data were valid and 4 sets were invalid. The invalid data were as follows: M—M_2_ (1 invalid data), M—M_3_ (1 invalid data), T—M_1_ (1 invalid data), and T—M_2_ (1 invalid data).

**Table 3 microorganisms-13-02802-t003:** Experimental scheme for *S. pasteurii* mineralization deposition.

Names	Ca^2+^ Concentration			Urea Concentration		Bacterial Suspension			Mineralization			
	Experiments	M	mL	M	mL	G	OD_600_	mL	Addition Order	Reaction	Filtration	Drying
Ca^2+^ concentration	15	0, 0.2, 0.4, 0.6, 0.7, 0.8, 0.9, 1, 1.1, 1.2, 1.3, 1.4, 1.6, 1.8, 2	50	1	50	3	0.5	5	Co-add	Quiescent	Centrifuge	Oven
	5	0.7	200	0.7	200	3	0.5	20				
		0.9		0.9								
		1		1								
		1.1		1.1								
		1.3		1.3								
Addition order and reaction	7	0.5, 0.8, 0.9, 1, 1.1, 1.2, 1.5	50	1	50	3	0.5	5	Premix	Shaking	Centrifuge	Oven
									Post-add			
									Premix	Quiescent		
									Post-add			
Filtration and drying	5	0.5, 0.9, 1, 1.1, 1.5	50	1	50	3	0.5	5	Co-add	Quiescent	Centrifuge	Room
												Oven
											Undisturbed	Room
												Oven

Addition order represents the addition order of *S. pasteurii* bacterial suspension, urea concentration, and Ca^2+^ concentration. When mixing the liquid, a KQ250DE numerically controlled ultrasonic cleaner was used to perform shaking for 2–3 min to ensure the liquid was well mixed. Premix represents a process of premixing the *S. pasteurii* bacterial suspension and urea, and then adding Ca^2+^. Co-add represents a process of mixing the *S. pasteurii* bacterial suspension, urea, and Ca^2+^ together. Post-add represents a process of premixing Ca^2+^ and urea, and then adding the solution to an *S. pasteurii* bacterial suspension. Reaction represents the reaction condition of the *S. pasteurii* bacterial suspension, the urea concentration, and the Ca^2+^ concentration. Shaking represents the process of achieving a mixture reaction using the orbital shaking condition. The culture temperature was set to 30 °C and oscillation speed was set to 170 r/rpm for 24 h. Quiescent refers to the mixture reaction without the orbital shaking condition. Filtration represents the filtration method for output. Centrifuge represents the filtration method with centrifugation. For centrifugation, the HC-3018 centrifuge is used for 3 min. It can be used again when the sample is still turbid until the solid and liquid are completely separated. Undisturbed represents the filtration method without centrifugation, leaving the mixture undisturbed. Drying represents the drying method used to achieve outputs. Oven represents the low-temperature drying method with an oven. For low-temperature drying, the oven temperature is set to 30 °C in 24 h. Room represents the room-temperature drying method in the shade without an oven.

**Table 4 microorganisms-13-02802-t004:** Feature importance weights of random forest.

Characteristics of X	No Interaction			Interaction with U			Difference	
	Importance	Ranked Importance	SD	Importance	Ranked Importance	SD	Importance	Ranked Importance
CT	0.601	1.517	0.067	0.568	1.134	0.067	↓5.49%	↓25.25%
IV	0.073	0.255	0.008	0.058	0.177	0.003	↓20.55%	↓30.59%
M	0.047	0.21	0.003	0.04	0.097	0.003	↓14.89%	↓53.81%
IA	0.077	0.208	0.023	0.076	0.285	0.026	↓1.3%	↑37.02%
pH	0.135	0.173	0.007	0.096	0.092	0.007	↓28.89%	↓46.82%
T	0.016	0.036	0.004	0.005	0.006	0.001	↓68.75%	↓83.33%
B	0.019	0.028	0.002	0.025	0.029	0.003	↑31.58%	↑3.57%
G	0.016	0.027	0.001	0.011	0.02	0.001	↓31.25%	↓25.93%
DO	0.015	0.008	0	0.012	0.009	0	↓20%	↑12.5%
U	—	—	—	0.108	0.267	0.023	—	—
**Characteristics** **of U**	**No interaction**			**Interaction with X**			**Difference**	
	**Importance**	**Ranked** **importance**	**SD**	**Importance**	**Ranked** **importance**	**SD**	**Importance**	**Ranked** **importance**
CT	0.276	1.027	0.017	0.122	0.236	0.006	↓55.8%	↓77.02%
IA	0.081	0.093	0.003	0.101	0.213	0.007	↑24.69%	↑129.03%
B	0.115	0.235	0.025	0.076	0.177	0.013	↓33.91%	↓24.68%
pH	0.349	0.299	0.02	0.062	0.094	0.007	↓82.23%	↓68.56%
IV	0.041	0.076	0.003	0.041	0.074	0.002	↓0.18%	↓2.63%
M	0.111	0.26	0.024	—	—	—	—	—
T	0.018	0.024	0.002	—	—	—	—	—
G	0.008	0.006	0	—	—	—	—	—
X	—	—	—	0.598	1.016	0.014	—	—
**Characteristics of u**	**No interaction**			— *				
	**Importance**	**Ranked** **importance**	**SD**					
t	0.269	1.341	0.137	—	—	—	—	—
IA	0.441	1.234	0.154	—	—	—	—	—
pH	0.214	0.916	0.043	—	—	—	—	—
B	0.033	0.094	0.009	—	—	—	—	—
IV	0.016	0.031	0.004	—	—	—	—	—
M	0.008	0.015	0.001	—	—	—	—	—
T	0.015	0.012	0.001	—	—	—	—	—
G	0.004	0.004	0	—	—	—	—	—

B refers to the batch of *S. pasteurii*. SD refers to standard deviation. * The variables X and U that might cause causal confusion. Therefore, when considering u, both of them were excluded. The down arrow represents the value of the interaction with U is lower compared to that without interaction. The up arrow represents the value of the interaction with U is higher compared to that without interaction.

**Table 5 microorganisms-13-02802-t005:** Weights table of key features and high-importance features of *S. pasteurii*.

Name	X				U				u			
	4 h	8 h	12 h	24 h	4 h	8 h	12 h	24 h	4 h	8 h	12 h	24 h
pH-0 h	0.189	0.498	0.42	—	—	1.566	0.32	—	0.58	0.58	0.112	0.147
pH-	1.055	—	0.116	—	0.532	—	—	0.109	0.316	0.282	—	0.28
X-0 h	0.154	1.477	1.317	—	0.263	—	—	1.127	—	—	—	0.236
X-	—	—	—	—	0.589	0.242	0.248	0.259	—	—	—	—
IV	—	—	0.113	0.486	—	—	0.33	—	—	—	0.882	0.355
G	—	—	—	0.322	—	—	—	0.133	—	—	—	0.279
t	—	—	—	0.122	—	—	0.125	—	—	—	—	—

**Table 6 microorganisms-13-02802-t006:** Statistical table of optimal culture conditions for different prediction targets of *S. pasteurii*.

M	Cultivation Conditions	ACt (h)	T (℃)	pH	DO (r/min)	IA (h)	IV (%)
M_1_	X	105	30	7.25	170	48	3
	U	27	32	8		16	1
M_2_	X	105	37	7	—	—	—
	U	54	30	9	—	—	—
M_3_	X	105	—	7	—	—	—
	U	27	—	8	—	—	—

“—” indicates that no experiment is conducted or no conclusion is drawn.

**Table 7 microorganisms-13-02802-t007:** Results of quantitative XRD analysis.

Name	B	C	D	I	II	III	IV
Addition	Post-add	Premix	Post-add	Co-add	Co-add	Co-add	Co-add
Reaction	Shaking	Quiescent	Quiescent	Quiescent	Quiescent	Quiescent	Quiescent
Filtration	Centrifuge	Centrifuge	Centrifuge	Undisturbed	Centrifuge	Undisturbed	Centrifuge
Drying	Oven	Oven	Oven	Room	Room	Oven	Oven
Calcite yield	5.6%	100%	5.6%	7.0%	7.9%	6.1%	5.0%

Calcite yield represents the yield of calcite in CaCO_3_.

**Table 8 microorganisms-13-02802-t008:** Effects of different filtration and drying methods on mineralization quality and the yield of CaCO_3_ in output of *S. pasteurii* after 24 h.

Calcium Acetate	I		II		III		IV		Theoretical Mass (g)
	Net Mass (g)	Caco_3_ Yield	Net Mass (g)	CaCO_3_ Yield	Net Mass (g)	CaCO_3_ Yield	Net Mass (g)	CaCO_3_ Yield	
0.5	1.616	0.65	2.072	0.83	2.125	0.85	1.892	0.76	2.5
0.9	2.794	0.62	1.762	0.39	2.172	0.48	2.72	0.6	4.5
1	3.031	0.61	3.14	0.63	3.38	0.68	2.664	0.53	5
1.1	3.275	0.66	2.905	0.58	3.701	0.74	3.542	0.71	5
1.5	4.405	0.88	4.066	0.81	1.903	0.38	3.609	0.72	5
Total	15.12	0.687	13.945	0.634	13.281	0.604	14.427	0.656	22

CaCO_3_ yield represents the yield of CaCO_3_ in output.

## Data Availability

The original contributions presented in the study are included in the article. Further inquiries can be directed to the corresponding author.

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
