# Peer review of "Research on MICP Restoration Technology for Earthen City Walls Damaged by Primary Vegetation Capping in China"

_microorganisms, 2025, doi:10.3390/microorganisms13122802_

Round 1

Reviewer 1 Report

Comments and Suggestions for Authors

Kind regards,

I hereby submit my suggestions for the authors, aiming to enhance the manuscript, which presents an interesting application of biogenic calcium carbonate technology.

Abstract

In general, it is recommended to avoid the "laboratory report" style used in the description of certain aspects and to place greater emphasis on the significance of the findings for sustainable heritage restoration.

Lines 12–14: Convoluted sentence, unclear at the beginning. Lacks context about the phenomenon under study.

Lines 14–15: The expression “To the end” is ambiguous. Suggested revision: "This study proposes the use of MICP as a biotechnological approach to reinforce root-soil composites and mitigate damage."

Introduction

Lines 37–41: “The difficulties lie in the fact as follows” is not natural in academic English. Suggested revision: "These sites face two main challenges: (i) continuous exposure to environmental fluctuations, which accelerates deterioration; and (ii) their large scale, which makes artificial control methods unfeasible."

Line 44: Replace "disease" with "deterioration mechanism" or "degradation phenomenon".

Lines 56–57: It is recommended to include a conceptual figure or cite an existing model of "wet migration" to strengthen and improve the understanding of this section.

Lines 60–63: The sentence should conclude by introducing microbial methods as environmentally friendly alternatives.

Materials and Methods

2.1. Study site: If possible, it is suggested to include the following aspects: (i) the sampling date, and (ii) whether permission or authorization was obtained to intervene in a heritage site, specifying the agreement or registration under which it was granted.

2.2 Experimental design: It is recommended that the authors include a table of abbreviations to improve the readability and flow of the text. In addition, the number of replicates performed for each assay should be included.

Tables 2 and 3: They present a large amount of information, which can make reading somewhat confusing. It is recommended to add a complete explanatory caption below the tables. Briefly describe the addition strategies (e.g., "Pre-addition: bacteria added before the calcium source; Post-addition: added after," etc.).

2.4 Study methods (RF-RFE-CV statistical model): Indicate which software or library was used (e.g., Python with scikit-learn, R, etc.). It is also important to specify the number of iterations, the dataset size, and whether overfitting/underfitting criteria were applied.

Results

Line 259: Incorrect term “high-base”; replace with “high-cardinality”.

3.1.2. Cross-sectional analysis: Please clarify the following: were new models generated at each time point? Was a single matrix used? 

Figures 6a to 6c, Figure 7, and Figure 10.: Remove the zeros from the time labels: "0, 24, 48, 72", and label the axis as "Time (h)". Please review and correct it where necessary.

3.2. Breeding conditions and 3.3. Mineralization conditions: It is important to indicate whether a statistical analysis of differences between treatments was conducted. Additionally, it is essential to specify the number of replicates per assay, as the graphs should include error bars representing the variation at each data point.

Discussion

4.1. Pasteurii features of growth and urease production: How do these findings relate to the bacterial growth phase? Are there implications for scaling up or field formulation? Are there previous studies with which these results align or diverge? Cite them briefly. Are these experimental conditions reproducible in real soils with temperature and pH fluctuations?

4.2. Pasteurii features of mineralization: What explains the negative effect of high Ca²⁺ concentrations? Is there formation of amorphous phases or blockage due to saturation? Can the "mix" or "post" strategy be applied under real conditions with surface irregularity or variable soil porosity? Has biofilm formation been considered as part of the nucleation mechanism in the treatments? This could be added as a hypothesis. Why does shade drying enhance performance? Is there evidence that it prevents phase transformation or loss of crystallinity?

Conclusions

(1) Parameter optimization: Indicate how this optimization may contribute to improving the effectiveness of MICP in real-world contexts.

(2) Scheme for improving urease activity: A clearer interpretation of the experimental conditions is recommended, particularly regarding their combined effect or the reasons why they enhance enzymatic activity. Also, consider projecting the potential for these conditions to be validated in real matrices (e.g., soil with roots, variable moisture). If they have already been validated, highlight the contribution of this study.

(3) Scheme for enhancing calcite yield: Indicate the intended future application (e.g., crack filling, surface consolidation, in situ treatment).

Author Response

Comments 1:

It is recommended to avoid the "laboratory report" style used in the description of certain aspects and to place greater emphasis on the significance of the findings for sustainable heritage restoration.

Response 1:

We thank the reviewer for this insightful suggestion. We have revised the manuscript to emphasize the broader implications of our findings for sustainable heritage restoration. 

Correction 1:

Line 131-135

This study designed a research plan that combined multi-factor experiments with mathematical models. The main objective was to systematically reveal the interaction of multiple factors on characteristics of growth, urease production, and mineralization in MICP process and their importance weights, and to construct a comprehensive structural understanding of the mineralization mechanism of S. pasteurii.

Comments 2:

Lines 12–14: Convoluted sentence, unclear at the beginning. Lacks context about the phenomenon under study.

Response 2:

We apologize for this oversight. We have rewritten the sentence to improve clarity and provide better context for the phenomenon under study. 

Correction 2:

Line 16-18

Based on the integration of MICP technology with plants offered advantages including soil solidification, erosion resistance, and resilience to dry-wet and freeze-thaw cycles, the application of MICP technology to the root-soil composite was proposed as a potential solution.

Comments 3:

Lines 14–15: The expression “To the end” is ambiguous. 

Response 3:

We apologize for this oversight. The ambiguous phrase "To the end" has been deleted.

Correction 3:

Line 16-18

Based on the integration of MICP technology with plants offered advantages including soil solidification, erosion resistance, and resilience to dry-wet and freeze-thaw cycles, the application of MICP technology to the root-soil composite was proposed as a potential solution.

Comments 4:

Lines 37–41: “The difficulties lie in the fact as follows” is not natural in academic English. 

Response 4:

We apologize for this oversight. We have rephrased the sentence to better align with academic English conventions. The revised text now reads: "These sites face two main challenges: "

Correction 4:

Line 44-46

These sites face two main challenges: (i) continuous exposure to environmental fluctuations, which accelerates deterioration; and (ii) their large scale, which makes artificial control methods unfeasible.

Comments 5:

Lines 56–57: It is recommended to include a conceptual figure or cite an existing model of "wet migration" to strengthen and improve the understanding of this section.

Response 5:

We thank the reviewer for this insightful suggestion. We have added a conceptual figure 1 from the literature to enhance clarity.

Comments 6:

Lines 60–63: The sentence should conclude by introducing microbial methods as environmentally friendly alternatives.

Response 6:

We thank the reviewer for this insightful suggestion. This sentence has been revised. At the end, it clearly introduces the microbial method as an environmental-friendly alternatives.

Correction 6:

Line 77

Microbial Induced Carbonate Precipitation (MICP) is widely observed in nature. There microbial mineralization is diverse. Wu et al. (2022) [11], Liu et al. (2018) [12], Wang et al. (2025) [13], Whiftin et al. (2007) [14], Al-Tabbaa et al. (2019) [15] and Simone et al. (2025) [16] proved that the cementitious materials precipitated by mineralization could be applied to stone, historical buildings, concrete, and sandy soil materials, featuring functions such as film protection, self-restoration, and environmental-friendly alternatives. In recent years, Daryono et al. (2024) [17], Deng et al. (2025) [18], Wang et al. (2024) [19], and Murugan et al. (2021) [20] started to explore the combined effect of MICP technology with plants from the perspective of soil synergistic reinforcement and environmental protection.

Comments 7:

2.1. Study site: If possible, it is suggested to include the following aspects: (i) the sampling date, and (ii) whether permission or authorization was obtained to intervene in a heritage site.

Response 7:

We apologize for this oversight. We have now included the sampling date and have given relevant support bureau on the heritage site. This information has been added to Section 2.1.

Correction 7:

Line 153-157

The sampling date was May 15, 2022. We didn’t obtain the permission document, but in 2023, we were in charge of Shanxi Provincial Cultural Relics Technology Program of China (2023KT15) organized by the Shanxi Provincial Cultural Relics Bureau, which clearly stipulated that the research scope was Western City wall in Jinyang Ancient City.

Comments 8:

2.2 Experimental design: It is recommended that the authors include a table of abbreviations to improve the readability and flow of the text. In addition, the number of replicates performed for each assay should be included.

Response 8:

We apologize for this oversight. We have included a table of abbreviations in Table 2 and Table 3 to improve readability. The number of replicates for each assay has been explicitly stated in the revised section of table 2.

Correction 8:

Line 191-200

Experiment names explanation: M refers to the three different culture media. T refers to the culture temperature of media. pH refers to the pH value. DO refers to dissolved oxygen. IA refers to inoculation age of S. pasteurii. IV refers to inoculation volume of S. pasteurii. t refers to the cultivation time of mixture. G refers to the cultivation generation of S. pasteurii.

Symbol explanation: “/” indicates a single-factor element, and “—" indicates that the factor is not measured.

Experimental group description: The optimization design involved a total of 115 experiments, among which 111 sets of data are valid and 4 sets were invalid. The invalid data are as follows: M—M2 (1 invalid data), M—M3 (1 invalid data), T—M1 (1 invalid data), and T—M2 (1 invalid data).

Line 202-218

Addition represents addition order of S. pasteurii. When mixing the liquid, a KQ250DE numerically controlled ultrasonic cleaner is used to shake for 2-3 minutes to ensure the liquid is well mixed. Premix represents to premix the liquid of S. pasteurii and urea, and then to add Ca2+ concentration. Co-add represents to mix the liquid of S. pasteurii, urea, and Ca2+ concentration together. Post-add represents to premix Ca2+ concentration and urea, and then to add the liquid of S. pasteurii.

Reaction represents the reaction condition of S. pasteurii, urea, and Ca2+ concentration. Shaking represents the mixture reaction with orbital shaking condition. The culture temperature was set to 30℃ and oscillation speed was set to 170r/rpm for 24 hours. Quiescent represents the mixture reaction without orbital shaking condition.

Filtration represents the filtration method for output. Centifuge represents the filtration method with centrifugation. For centrifugation, the HC-3018 centrifuge is used into 3 minutes. It can be used again when the sample is still turbid, until the solid and liquid is completely separated. Undisturbed represents the filtration method without centrifugation, leaving mixture undisturbed.

Drying represents the drying method for output. Oven represents the low-temperature drying method with oven. For low-temperature drying, the oven temperature was set to 30℃ in 24 hours. Room represents the room temperature drying method in the shade without oven.

Comments 9:

Tables 2 and 3: They present a large amount of information, which can make reading somewhat confusing. It is recommended to add a complete explanatory caption below the tables. Briefly describe the addition strategies (e.g., "Pre-addition: bacteria added before the calcium source; Post-addition: added after," etc.).

Response 9:

We apologize for this oversight. We have added comprehensive explanatory captions to Tables 2 and 3. The captions now clearly describe the addition strategies.

Comments 10:

2.4 Study methods (RF-RFE-CV statistical model): Indicate which software or library was used (e.g., Python with scikit-learn, R, etc.). It is also important to specify the number of iterations, the dataset size, and whether overfitting/underfitting criteria were applied.

Response 10:

We apologize for this oversight. We have specified that the RF-RFE-CV model was implemented using Python with the scikit-learn library. Details on the number of iterations, dataset size, and measures taken to avoid overfitting/underfitting have been added to this section.

Correction 10:

Line 281-290

Based on the fact that the S. pasteurii strain exhibited significant individual variability, leading to considerable variation in the results of repeated experiments. Moreover, the influence of each feature on the prediction target for the S. pasteurii strain varied greatly, and there was a high likelihood of multi-collinearity among the features. This paper attempted to explore how to use large amounts of data to find the physiological and mineralization characteristics of S. pasteurii and provided a reference for future S. pasteurii breeding work. A total of 111 items with million-level data were included in the comprehensive statistical experimental group. This feature analysis established a Random Forest - Recursive Feature Elimination - Cross Validation model (RF-RFE-CV). And the ultimate goal was to obtain the optimal feature set, to master the feature importance weights, and significantly to improve the model performance. This paper used Spyder module in the software of Anacondas in python language with scikit-learn library, to implement an optimized multi-objective feature selection analysis model. This model effectively prevented the problem of underfitting by setting a minimum feature selection threshold of at least three important features, using a median imputation strategy to handle missing values, and applying an outlier truncation method to maintain data integrity. It systematically prevented the risk of overfitting through a combination of measures such as limiting the tree number of RF to 100, using ECV to automatically determine the optimal feature subset, implementing 5-fold cross-validation to evaluate the model's generalization ability, and conducting three repeated permutation importance tests to assess feature stability.

Comments 11:

Line 259: Incorrect term “high-base”; replace with “high-cardinality”.

Response 11:

We apologize for the error. The term "high-base" has been corrected to "high-cardinality" as suggested.

Correction 11:

Line 309

That was because pH, IA, U, etc., had a large number of values, which were typical high-cardinality characteristics.

Comments 12:

3.1.2. Cross-sectional analysis: Please clarify the following: were new models generated at each time point? Was a single matrix used? 

Response 12:

We apologize for this oversight. We have clarified the methodology: some models were applied across all time points using multi-matrix. 

Correction 12:

Line 340-349

On the basis of above-mentioned feature parameters, the initial pH, initial X, and initial U were added to calculate the feature importance of predicted target feature parameters of S. pasteurii at different time points. During the analysis, independent prediction models were constructed for the target variables at each time point. Different feature selection strategies were adopted for various biological indicators. When predicting X, the combined influence of multiple factors and U was considered. When predicting U, the combined influence of multiple factors and X was considered. However, when predicting u, the variables X and U were excluded, and only the influence of multiple factors was considered. This multi-matrix modeling approach could more accurately capture the specific influencing factors of different target variables, avoiding the information redundancy or causal inversion that may result from a single matrix, and enhancing the interpretability and predictive accuracy of the model.

Comments 13:

Figures 6a to 6c, Figure 7, and Figure 10.: Remove the zeros from the time labels: "0, 24, 48, 72", and label the axis as "Time (h)". Please review and correct it where necessary.

Response 13:

We apologize for this oversight. We have revised all relevant figures as recommended: the zeros have been removed from time labels and the x-axis is now consistently labeled as "Time (h)".

Comments 14:

3.2. Breeding conditions and 3.3. Mineralization conditions:

It is important to indicate whether a statistical analysis of differences between treatments was conducted. Additionally, it is essential to specify the number of replicates per assay, as the graphs should include error bars representing the variation at each data point.

Response 14:

Thank you for your valuable suggestions. The statistical analysis of differences between treatments was shown in Table 2. Due to the constraints of experimental conditions and time, it is difficult for us to conduct supplementary experiments in this revision. Although this study was based on only 111 sets of experimental conditions, the total amount of raw data reached the million-level, meeting the basic requirements of big data analysis.

However, we have discussed the limitations of the study in the discussion section and highlighted the contribution of this paper.

Your suggestions will be an important part of our future work. Thank you again for your help with the manuscript.

Correction 14:

Line 540-545

Although this study was based on only 111 sets of experimental conditions, the total amount of raw data reached the million-level, meeting the basic requirements of big data analysis. The core contribution of this paper lay in proposing and validating a Random Forest methodological framework to provide references for subsequent research to achieve more accurate and interpretable analysis and prediction of mineralization mechanisms.

Comments 15:

4.1. Pasteurii features of growth and urease production: How do these findings relate to the bacterial growth phase?

Are there implications for scaling up or field formulation?

Are there previous studies with which these results align or diverge? 

Are these experimental conditions reproducible in real soils with temperature and pH fluctuations?

Response 15:

Thank you for your valuable suggestions. We have revised the discussion in 4.1 to enlarge discussion for this paper.

Correction 15:

4.1. S. pasteurii features of urease production

(3) Comprehensive analysis

It was known that U was a core functional indicator for predicting and evaluating the mineralization efficiency of S. pasteurii. Longitudinal analysis (Table 4) showed that when the characteristic parameters of U interacted with X, the number of characteristic parameters reduced from 8 to 6, and X (1.016) became a key characteristic of S. pasteurii, which was far more important than other characteristic parameters. Ignoring the continuity of t, the cross-sectional data (Table 5) of S. pasteurii showed that the weights of X-4h and X-0h at 4-hour were 0.589 and 0.263 respectively, the weight of X-8h at 8-hour was 0.242, the weight of X-12h at 12-hour was 0.248, and the weights of X-0h and X-24h at 24-hour were 1.127 and 0.259 respectively. X was high importance feature at multiple time points such as at 4-hour, 8-hour, 12-hour and 24-hour, and even at 24-hour, X-0h became a key feature. That was, the best response time was at around 24-hour, which proved that urease activity was highly dependent on the initial X in the later stage of culture, and the growth potential in the early stage was fully reflected at this stage. In other words, X was a key driving factor for characterizing U, and its inclusion in U prediction model could reduce the number of characteristic parameters and improve the explanatory power of each characteristic parameter. Its importance was confirmed in both longitudinal and cross-sectional data analysis. It can be seen that U was closely related to the initial pH value and the number of bacteria. A higher initial X could prompt S. pasteurii to enter the rapid proliferation period faster, which would promote the overall increase of X and U. The above conclusion was in line with the general law of bacterial growth.

Longitudinal analysis (Table 4) indicated that t (1.517) was the most critical factor influencing X, with the ranking of highly influential factors followed the rule of IV (0.255) > M (0.21) > IA (0.208) > pH (0.173). Cross-sectional analysis revealed distinct temporal patterns in feature importance. pH-4h and pH-0h at 4-hour exhibited weights of 1.055 and 0.189, respectively, while X-0h had a weight of 0.154. pH-0h value at 8-hour decreased to 0.498, whereas X-0h increased significantly to 1.477. pH-0h and pH-12h at 12-hour were weighted at 0.42 and 0.116, respectively, with X-0h remaining high at 1.317. The influence of pH and X-0h at 24-hour diminished, while IA and G emerged as dominant factors, with weights of 0.486 and 0.322, respectively. Redundancy analysis of bacterial concentration-related features demonstrated a redundancy weight of 1.052 between t and pH, and 0.917 between IA and IV, suggesting potential multi-collinearity issues. In the growth curve of S. pasteurii, pH was identified as the key determinant during the lag phase, with pH-4h and pH-0h at 4-hour weighted at 1.055 and 0.189, respectively—indicating strong regulation by the initial culture medium environment. The results of pH experiment demonstrated that maintaining pH within the range of 7.25 - 9 could effectively shorten the duration of the lag phase, further validating the decisive influence of pH on X during the lag phase. During the exponential growth phase, X-0h at 8-hour and 12-hour became the predominant factor, with weights of 1.477 and 1.317, aligning well with the expected growth kinetics of S. pasteurii during logarithmic proliferation. This highlighted the decisive impact of inoculation strategy on U synthesis in this phase. In contrast, during the stationary or deceleration phase (24 h), influencing factors became more diverse, with weights of 0.486 and 0.322 on IV and G increasing in relative importance, reflecting a complex regulatory stage of bacterial growth. This shift was likely attributable to nutrient depletion and the accumulation of metabolic by-products in later culture stages, which collectively suppress growth rates. These findings were in close agreement with the established growth profile of S. pasteurii under nutrient-limited conditions and corroborate previous reports by Wang (2009) [23] and Wei (2023) [53], thereby enhancing the reliability and generalizability of the conclusions.

In addition to X, the ranking of high-importance features for U in the longitudinal analysis followed the rules of t (0.236) > IA (0.213) > B (0.177). In the cross-sectional analysis, U exhibited pH-4h at 4-hour showing a weight of 0.532. pH-0h at 8-hour reached the highest observed weight of 1.566. pH-0h at 12-hour registered 0.32 and pH-24h at 24-hour contributed a weight of 0.109. These results indicated that pH was a key determinant of S. pasteurii in U, particularly during the early exponential growth phase. The peak importance of pH-0h at 8-hour underscored its critical role in modulating U, further validating pH as a major influencing factor in cross-sectional analysis.

u served as a core quality indicator for assessing U, widely utilized in the screening of high-efficiency strains and the optimization of culture conditions. Longitudinal analysis revealed that the key determinants of u were ranked in descending order of culture time (1.341) > inoculation age (1.234), with pH emerging as a highly influential factor (0.916). Cross-sectional analysis demonstrated distinct temporal patterns in regulatory influences. During the early growth phase, u in S. pasteurii was predominantly governed by pH conditions. pH-0h exhibited a consistent weight of 0.580 at 4-hour and 8-hour, while pH-4h and pH-8h registered weights of 0.316 and 0.282, respectively. As cultivation progresses at 12-hour and 24-hour, the influence of IV increased markedly, with importance weights of 0.882 and 0.355. Concurrently, the role of pH diminished—pH-0h at 12-hour dropped to 0.112, and pH-24h and pH-0h at 24-hour contributed only 0.280 and 0.147, respectively. Redundancy analysis of u revealed a redundancy weight of 0.956 between IA and IV, suggesting a significant degree of multi-collinearity. Cross-sectional evaluation identified pH and IV as primary high-importance features. The significance of pH was consistently validated across multiple time points, whereas the impact of IV became prominent after 12-hour, highlighting its growing regulatory role in the mid-to-late culture phase.

  1. Conclusions

(1) Parameter optimization.

The U of S. pasteurii was primarily regulated by three key parameters with X, pH, and u, which required stage-specific dynamic control throughout the growth cycle. During the optimization of U, X consistently emerged as a high-importance feature across multiple time points, with peak effectiveness observed at 24-hour (1.127). However, the regulatory factors of X varied across growth phases. Across the lag, rapid growth, and slow growth phases, the dominant influence on X progressively shifted from pH to X-0h, and ultimately to a broader set of regulatory factors, reflecting increasingly complex physiological and environmental interactions. pH remained a highly influential parameter across several time points, exhibiting maximum impact at around 8-hour (1.566). With u, the significant factors were pH and IV. pH exerted a pronounced influence during the early cultivation stage, whereas IV gained increasing importance after 12-hour, underscoring its role in sustaining metabolic activity in mid-to-late culture phases. The above conclusion would provide important experimental data references in actual soil where temperature and pH values fluctuate for scaling up or field formulation.

Comments 16:

4.2. Pasteurii features of mineralization: What explains the negative effect of high Ca²⁺concentrations? Is there formation of amorphous phases or blockage due to saturation?

Can the "mix" or "post" strategy be applied under real conditions with surface irregularity or variable soil porosity?

Has biofilm formation been considered as part of the nucleation mechanism in the treatments? This could be added as a hypothesis. Why does shade drying enhance performance? Is there evidence that it prevents phase transformation or loss of crystallinity?

Response 16:

Thank you for your valuable suggestions. We have revised the discussion in 4.2 to enlarge discussion for this paper.

Correction 16:

4.2. S. pasteurii features of mineralization

 (1) The yield of CaCO3 in output

The research on Ca2+ concentration confirmed that the yield of CaCO3 in output was higher when the same low concentration of calcium source and urea solution were used. Ghosh et al. (2005) [59] found that an excessively high enzymatic hydrolysis rate was not conducive to enhanced mineralization, and a slow and stable rate of calcite mineralization deposition was more beneficial for the enhancement effect of carbonate mineralization bacteria. Jamal et al. (2023) [60] and Xu et al. (2024) [61] indicated that this broad adaptability couldn’t be directly translated into the optimal efficiency of the MICP process. Blindly pursuing high urease activity might be counterproductive. When the urea solution concentration was 1M, the net mass and the yield of CaCO3 in output were higher when the Ca2+ concentration was 1.3M. It can be seen that when the urease activity of S. pasteurii was constant, the ratio of urea to Ca2+ concentration at 1M:1.3M was more conducive to the formation of high-quality net mass and the yield of CaCO3 in output.

As shown in table 8, with the filtration methods, centrifugal filtration was faster than undisturbed filtration, but the yield of CaCO3 in output followed undisturbed filtration > centrifugal filtration, with an error of 5.2% to 5.4%. With the drying method, oven drying was quicker than room temperature drying in the shade, but the yield of CaCO3 in output followed room temperature drying in the shade > oven drying, with an error of 3% to 3.1%. However, based on the fact that centrifugal filtration could quickly and thoroughly collect all solid precipitates onto the filter paper, in many cases, considering the time-saving aspect, centrifugal filtration and oven drying were often adopted to speed up the process, with an error of about 8.3%.

Therefore, to maximize the yield of CaCO3 in output, it was recommended to follow the condition of the ratio of urea to Ca2+ concentration at 1M:1.3M, undisturbed filtration and room temperature drying in the shade.

(2) The yield of calcite in CaCO3

As shown in Figure 11 (b) - (d), the crystals of the mineralization products in group B were mostly needle-like, while those in group D were stacked to form disc-shaped structures ranging from 10 to 20μm. Group C presented granular and blocky forms, with some crystals bonded together, and the crystal size ranged from 2 to 15μm. This indicated that the addition order of S. pasteurii had an impact on the appearance of the mineralization products. By comparing the data of Group B, Group C and Group D in Table 7 and the samples in Figure 11 (b) - (d) and Figure 9 (e), different addition methods of S. pasteurii had different impacts on the crystal form of the mineralization product. The premix method was most conducive to the yield of calcite in CaCO3, which could reach 100%.

As shown in the relevant data of Group I, Group II, Group III, and Group IV in Table 7, the form of centrifugal filtration and room temperature drying in the shade was more likely to form calcite. Among the filtration methods, the yield of calcite in CaCO3 followed the rules of centrifugal filtration > undisturbed filtration, and the error range was from 0.9% to 1.1%. Among the drying methods, the yield of calcite in CaCO3 followed the rules of room temperature drying in the shade > oven drying, and the error range was from 1.8% to 2%.

Therefore, to maximize the yield of calcite in CaCO3, it was recommended to follow the condition of premix method of S. pasteurii, quiescent reaction, centrifugal filtration, and room temperature drying in the shade.

(3) Comprehensive analysis

A comparison of the recommended conditions for the yield of CaCO3 in output and the yield of calcite in CaCO3 revealed that the conflicting condition was the filtration method. The yield of CaCO3 in output needed undisturbed filtration, while the yield of calcite in CaCO3 needed centrifugal filtration.

The lower yield of CaCO₃ in output observed in centrifugal filtration may be attributed to the loss of tiny particles during the post-centrifugation process. The high gravity during centrifugation effectively precipitated larger and well-crystallized particles. However, it also caused some tiny CaCO₃ precursors to be lost along with the supernatant when poured on the filter paper. The higher yield of calcite in CaCO₃ observed in centrifugal filtration may be attributed to the fact that calcite particles were larger in size compared to the particles of aragonite and vaterite, making them more likely to remain on the filter paper.

Among the mineralization products obtained from Group I, II, III and IV, the yield of calcite in outpt were 4.27%, 4.98%, 4.15%, and 2.65%, respectively. Under the same condition of room temperature drying in the shade, the difference in calcite content between undisturbed filtration in Group I and centrifugal filtration in Group II was only 0.71%, indicating no significant difference in crystal form selectivity between them. Meanwhile, the yield of CaCO₃ in output with the condition of undisturbed filtration was higher than that of centrifugal filtration in 5.2%–5.4%, demonstrating a clear advantage in output. Therefore, considering the stability of output, the condition of undisturbed filtration was more appropriate.

In summary, to maximize the yield of CaCO3 in output and the yield of calcite in CaCO3, it was recommended to follow the condition of the ratio of urea to Ca2+ concentration at 1M:1.3M, premix method of S. pasteurii, quiescent reaction, undisturbed filtration, and room temperature drying in the shade.

Comments 17:

Conclusions

(1) Parameter optimization: Indicate how this optimization may contribute to improving the effectiveness of MICP in real-world contexts.

(2) Scheme for improving urease activity: A clearer interpretation of the experimental conditions is recommended, particularly regarding their combined effect or the reasons why they enhance enzymatic activity. Also, consider projecting the potential for these conditions to be validated in real matrices (e.g., soil with roots, variable moisture). If they have already been validated, highlight the contribution of this study.

(3) Scheme for enhancing calcite yield: Indicate the intended future application (e.g., crack filling, surface consolidation, in situ treatment).

Response 17:

Thank you for your valuable suggestions. We have revised the Conclusions section to address these points: (1) We stated how parameter optimization could improve MICP effectiveness in real-world contexts. (2) We provided a clearer interpretation of the experimental conditions and their combined effect on urease activity. (3) It is difficult for us to add relevant experiment to indicate the intended future application. However, we highlighted the study's contribution.

Correction 17:

  1. Conclusions

 (1) Parameter optimization.

The U of S. pasteurii was primarily regulated by three key parameters with X, pH, and u, which required stage-specific dynamic control throughout the growth cycle. During the optimization of U, X consistently emerged as a high-importance feature across multiple time points, with peak effectiveness observed at 24-hour (1.127). However, the regulatory factors of X varied across growth phases. Across the lag, rapid growth, and slow growth phases, the dominant influence on X progressively shifted from pH to X-0h, and ultimately to a broader set of regulatory factors, reflecting increasingly complex physiological and environmental interactions. pH remained a highly influential parameter across several time points, exhibiting maximum impact at around 8-hour (1.566). With u, the significant factors were pH and IV. pH exerted a pronounced influence during the early cultivation stage, whereas IV gained increasing importance after 12-hour, underscoring its role in sustaining metabolic activity in mid-to-late culture phases. The above conclusion would provide important experimental data references in actual soil where temperature and pH values fluctuate for scaling up or field formulation.

(2) Scheme for improving urease activity.

To obtain higher urease activity, it was recommended to prioritize the use of Medium 220. CASO AGAR (Merck 105458) as the culture medium for S. pasteurii. The activation culture time was 27-hour. The inoculation age was 16-hour. The inoculation amount was 1%. The culture temperature was set at 32°C. The initial pH value was 8. And oscillation speed was 170 r/min.

(3) Scheme for enhancing calcite yield.

When the urease activity of Bacillus was constant, a ratio of 1M calcium source to 1.3M urea was more conducive to the formation of high-quality CaCO3 net mass and CaCO3 yield. To achieve a higher CaCO3 yield and calcite yield, it was recommended to choose the technology of pre-addition on S. pasteurii, static nucleation, undisturbed filtration, and shade drying.

This study was based on systematic experimental data and introduced the random forest algorithm for big data analysis. It not only achieved a comparative analysis of the influence of single factors but also revealed the importance weights of multiple factors on MICP from a macro perspective. The aim of this data analysis method was to help readers form a comprehensive and structural understanding of the mineralization influence mechanism.

Reviewer 2 Report

Comments and Suggestions for Authors

Dear editor and dear authors,

I have read the paper entitled “Research on MICP restoration technology for earthen city wall damaged by primary vegetation capping in China”. The paper covers an interesting topic about application of MICP technology for wall protection. The authors investigated key factors and parameters in the mineralization process of Sporosarcina pasteurii to define the optimal pathway for enhancing urease activity and calcite yield. The paper is precise and detailed within the scope of the subject it deals with and could be valuable to the scientific community.

However, there are some points that should be addressed before publication, listed below.

  • Firstly, the references are not listed according to the Instructions for Manuscript preparation. References are listed by author name, but they should be numbered according to the citation order.
  • The authors could add a short paragraph at the end of Introduction section stating the aim of the current study.
  • The Materials and methods section should be written in passive form, for example, “multiple samplings at different time points were conducted according to the experimental plan”, instead of ”should be conducted”. The authors should check the whole section and correct where needed.
  • The discussion needs to be expanded with the results of other authors who dealt with similar problems, commented on and related them to current results.
  • Bacteria names should be written in italic, the authors should check the whole manuscript and correct where needed.

Author Response

Comments 1:

The references are not listed according to the Instructions for Manuscript preparation. References are listed by author name, but they should be numbered according to the citation order.

Response 1:

We apologize for this oversight. We have reformatted the reference list according to the journal's guidelines, numbering all references in the order of their citation in the text.

Comments 2:

The authors could add a short paragraph at the end of Introduction section stating the aim of the current study.

Response 2:

Thank you for this suggestion. We added a clear paragraph at the end of the Introduction section that explicitly stated the specific aims and objectives of our current study.

Correction 2:

Line 131-135

This study designed a research plan that combined multi-factor experiments with mathematical models. The main objective was to systematically reveal the interaction of multiple factors in MICP process and their importance weights, and to construct a comprehensive structural understanding of the mineralization mechanism.

Comments 3:

The Materials and methods section should be written in passive form.

Response 3:

We apologize for this error. We have revised the entire Materials and methods section to consistently use passive voice, as recommended for academic writing in this discipline.

Correction 3:

  1. Materials and methods

2.1. Study site

The sampling site of Jinyang Ancient City is located in Jinyuan District, southwest of Taiyuan City, Shanxi Province, China. The vegetation distribution on the city wall was uneven overall. The western section of the wall (Figure 2) supported a mix of shrubs and small trees, while the eastern section was mainly covered with herbs and vines. Soil analysis from the site in Jinyang Ancient City (Shang et al., 2024) indicated that the predominant soil type was sandy loam, classified as non-collapsible loess. With increasing soil depth, the moisture content showed a corresponding rise. The primary soil salts consisted of CaSO4 and MgSO4, with pH values ranging from 7.1 to 7.45 and a specific gravity between 2.69 and 2.7.

2.2. Experimental design

There are numerous factors that affect the growth, reproduction, metabolism and mineralization of S. pasteurii, mainly including nutrients, culture conditions, inoculation conditions and mineralization conditions, etc. The main objective of this study was to provide high concentrations and high quality of S. pasteurii. In this section, X, U, u, and calcite mineralization yield were used as different prediction targets. The single-factor comparison experiments and orthogonal experiments were combined to investigate the variation rules of the prediction targets under different conditions respectively.

There were three common types of S. pasteurii culture media, respectively defined as M1, M2, and M3. The formulation was shown in Table 1. S. pasteurii was revived and activated by M1, with the culture time set at 27h, 54h, 79h, and 105h respectively. After the successful activation of the above strains, the following experimental work was completed.

The experimental design for the factors influencing the physiological characteristics of S. pasteurii was shown in Table 2. The experiment on the mineralization of CaCO3 was shown in Table 3. For the factor experiments, multiple samplings at different time points was conducted according to the experimental plan, and the measurement frequency also was planned. Parameters such as culture temperature, pH value, OD600 value, and the difference in electrical conductivity of S. pasteurii liquid between before and after 5-minute intervals were recorded. The remaining bacterial liquid in the conical flask during the culture process was kept under the original conditions for continued cultivation. Calcium acetate was chosen as the calcium source and was provided in solution form. S. pasteurii liquid with high urease was prepared for standby. According to the experimental requirements, the appropriate liquid mixing method and mineralization method was selected. The precision of the weighing balance was 0.0001g.

2.3. Characterization parameters

(1) Growth characteristics

The growth curve of S. pasteurii is a fluctuating curve plotted with time as the X-axis and S. pasteurii concentration as the Y-axis. It is a fundamental indicator for describing the growth and reproduction rules and is also the core for establishing a growth dynamics model of S. pasteurii. The absolute concentration in the model was approximately calculated by OD600 with Formula (3) (Wang, 2009).

                      (3)

where: X refers to absolute concentration of S. pasteurii, in 108CFU·mL-1; OD600 refers to optical density of the bacterial solution measured at 600nm, with no unit; n refers to dilution factor of samples, with no unit.

(2) Urease production characteristics

Urease activity is the ability to hydrolyze urea within a unit of time. Traditionally, urease activity of S. pasteurii was mostly determined qualitatively by colorimetry (Xiao, 2004), but its quantitative indicators cannot be obtained directly. Urease's decomposition rate of urea was characterized by measuring the rate of increase in electrical conductivity in the bacterial liquid. Its principle was that the increase in CO32- and NH4+concentrations during the decomposition of urea lead to an increase in the electrical conductivity of the bacterial liquid. Urease Activity curve of the S. pasteurii was the fluctuation curve of urease activity of the S. pasteurii culture liquid over time, showing the variation rule of urease activity. The calculation method was shown in Formula (4). Urease Activity Intensity refers to the ability of a unit of X to decompose urea within a unit of time. The calculation method was shown in Formula (5).

                   (4)

                                 (5)

where: U refers to urease activity, in µ?·(??·???)−1; f represents the rate of change in electrical conductivity,2 σ represents the initial electrical conductivity value, σ represents the electrical conductivity value after t minute, in m?·??−1; u refers to the intensity of urease activity, in µ?·mL·(CFU·??·???)−1; X refers to the absolute concentration of S. pasteurii liquid, in 108CFU·mL−1.

(3) Mineralization characteristics

The yield of mineralization products was measured by the index parameter method. The actual production of CaCO3 was measured on-site, and the CaCO3 yield was characterized by the ratio of the actual production mc of calcium carbonate to the theoretical production m0, denoted by η, as shown in Formula (6). The formula for calculating the theoretical production of CaCO3 was shown in Formula (7).

                     (6)

(7)

where: mc is the actual amount of CaCO3, mp is the mass of filter paper with sediment attached, mf is the mass of filter paper, m0 is the theoretical amount of CaCO3, m0-a is the theoretical amount of CaCO3 based on the mass of calcium source, m0-u is the theoretical amount of CaCO3 based on the mass of urea, ma is the mass of calcium source added before precipitation, mu is the mass of urea added before precipitation, in g; Ma is the relative molecular mass of calcium source, 158.17, Mc is the relative molecular mass of CaCO3, 100.09, Mu is the relative molecular mass of urea, 60.06, in g; η is the CaCO3 yield, in %.

Comments 4:

The discussion needs to be expanded with the results of other authors who dealt with similar problems, commented on and related them to current results.

Response 4:

We apologize for this oversight. We have significantly expanded the Discussion section to include more comprehensive comparisons with relevant previous studies. We now explicitly discuss how our results align with, contradict, or extend the findings of other researchers in this field.

Correction 4:

4.1. S. pasteurii features of urease production

 (3) Comprehensive analysis

It was known that U was a core functional indicator for predicting and evaluating the mineralization efficiency of S. pasteurii. Longitudinal analysis (Table 4) showed that when the characteristic parameters of U interacted with X, the number of characteristic parameters reduced from 8 to 6, and X (1.016) became a key characteristic of S. pasteurii, which was far more important than other characteristic parameters. Ignoring the continuity of t, the cross-sectional data (Table 5) of S. pasteurii showed that the weights of X-4h and X-0h at 4-hour were 0.589 and 0.263 respectively, the weight of X-8h at 8-hour was 0.242, the weight of X-12h at 12-hour was 0.248, and the weights of X-0h and X-24h at 24-hour were 1.127 and 0.259 respectively. X was high importance feature at multiple time points such as at 4-hour, 8-hour, 12-hour and 24-hour, and even at 24-hour, X-0h became a key feature. That was, the best response time was at around 24-hour, which proved that urease activity was highly dependent on the initial X in the later stage of culture, and the growth potential in the early stage was fully reflected at this stage. In other words, X was a key driving factor for characterizing U, and its inclusion in U prediction model could reduce the number of characteristic parameters and improve the explanatory power of each characteristic parameter. Its importance was confirmed in both longitudinal and cross-sectional data analysis. It can be seen that U was closely related to the initial pH value and the number of bacteria. A higher initial X could prompt S. pasteurii to enter the rapid proliferation period faster, which would promote the overall increase of X and U. The above conclusion was in line with the general law of bacterial growth.

Longitudinal analysis (Table 4) indicated that t (1.517) was the most critical factor influencing X, with the ranking of highly influential factors followed the rule of IV (0.255) > M (0.21) > IA (0.208) > pH (0.173). Cross-sectional analysis revealed distinct temporal patterns in feature importance. pH-4h and pH-0h at 4-hour exhibited weights of 1.055 and 0.189, respectively, while X-0h had a weight of 0.154. pH-0h value at 8-hour decreased to 0.498, whereas X-0h increased significantly to 1.477. pH-0h and pH-12h at 12-hour were weighted at 0.42 and 0.116, respectively, with X-0h remaining high at 1.317. The influence of pH and X-0h at 24-hour diminished, while IA and G emerged as dominant factors, with weights of 0.486 and 0.322, respectively. Redundancy analysis of bacterial concentration-related features demonstrated a redundancy weight of 1.052 between t and pH, and 0.917 between IA and IV, suggesting potential multi-collinearity issues. In the growth curve of S. pasteurii, pH was identified as the key determinant during the lag phase, with pH-4h and pH-0h at 4-hour weighted at 1.055 and 0.189, respectively—indicating strong regulation by the initial culture medium environment. The results of pH experiment demonstrated that maintaining pH within the range of 7.25 - 9 could effectively shorten the duration of the lag phase, further validating the decisive influence of pH on X during the lag phase. During the exponential growth phase, X-0h at 8-hour and 12-hour became the predominant factor, with weights of 1.477 and 1.317, aligning well with the expected growth kinetics of S. pasteurii during logarithmic proliferation. This highlighted the decisive impact of inoculation strategy on U synthesis in this phase. In contrast, during the stationary or deceleration phase (24 h), influencing factors became more diverse, with weights of 0.486 and 0.322 on IV and G increasing in relative importance, reflecting a complex regulatory stage of bacterial growth. This shift was likely attributable to nutrient depletion and the accumulation of metabolic by-products in later culture stages, which collectively suppress growth rates. These findings were in close agreement with the established growth profile of S. pasteurii under nutrient-limited conditions and corroborate previous reports by Wang (2009) [17] and Wei (2023) [51], thereby enhancing the reliability and generalizability of the conclusions.

In addition to X, the ranking of high-importance features for U in the longitudinal analysis followed the rules of t (0.236) > IA (0.213) > B (0.177). In the cross-sectional analysis, U exhibited pH-4h at 4-hour showing a weight of 0.532. pH-0h at 8-hour reached the highest observed weight of 1.566. pH-0h at 12-hour registeried 0.32 and pH-24h at 24-hour contributed a weight of 0.109. These results indicated that pH was a key determinant of S. pasteurii in U, particularly during the early exponential growth phase. The peak importance of pH-0h at 8-hour underscored its critical role in modulating U, further validating pH as a major influencing factor in cross-sectional analysis.

u served as a core quality indicator for assessing U, widely utilized in the screening of high-efficiency strains and the optimization of culture conditions. Longitudinal analysis revealed that the key determinants of u were ranked in descending order of culture time (1.341) > inoculation age (1.234), with pH emerging as a highly influential factor (0.916). Cross-sectional analysis demonstrated distinct temporal patterns in regulatory influences. During the early growth phase, u in S. pasteurii was predominantly governed by pH conditions. pH-0h exhibited a consistent weight of 0.580 at 4-hour and 8-hour, while pH-4h and pH-8h registered weights of 0.316 and 0.282, respectively. As cultivation progresses at 12-hour and 24-hour, the influence of IV increased markedly, with importance weights of 0.882 and 0.355. Concurrently, the role of pH diminished—pH-0h at 12-hour dropped to 0.112, and pH-24h and pH-0h at 24-hour contributed only 0.280 and 0.147, respectively. Redundancy analysis of u revealed a redundancy weight of 0.956 between IA and IV, suggesting a significant degree of multi-collinearity. Cross-sectional evaluation identified pH and IV as primary high-importance features. The significance of pH was consistently validated across multiple time points, whereas the impact of IV became prominent after 12-hour, highlighting its growing regulatory role in the mid-to-late culture phase.

4.2. S. pasteurii features of mineralization

The research on Ca2+ concentration confirmed that the yield of CaCO3 was higher when the same low concentration of calcium source and urea solution were used. Xiao (2004) [56] and Ghosh et al. (2005) [57] found that an excessively high enzymatic hydrolysis rate was not conducive to enhanced mineralization, and a slow and stable rate of calcite mineralization deposition was more beneficial for the enhancement effect of carbonate mineralization bacteria. Jamal et al. (2023) [58] and Xu et al. (2024) [59] indicated that this broad adaptability couldn’t be directly translated into the optimal efficiency of the MICP process. Blindly pursuing high urease activity might be counterproductive. When the urea solution concentration was 1M, the net mass and yield of CaCO3 were higher when the calcium source concentration was 1.3M. It can be seen that when the urease activity of Pseudomonas was constant, the ratio of calcium source to urea at 1M:1.3M was more conducive to the formation of high-quality net mass and yield of CaCO3. The experiment on the addition sequence of calcium source and the filtration and verification confirmed that the technology of pre-addition on S. pasteurii, static nucleation, undisturbed filtration, and shade drying could produce the highest yield of CaCO3, and also could produce the highest yield of calcite. The research on Ca2+ concentration confirmed that the yield of CaCO3 was higher when the same low concentration of calcium source and urea solution were used. When the urea solution concentration was 1M, the net mass and yield of CaCO3 were higher when the calcium source concentration was 1.3M. It can be seen that when the urease activity of Pseudomonas was constant, the ratio of calcium source to urea at 1M:1.3M was more conducive to the formation of high-quality net mass and yield of CaCO3. The experiment on the addition sequence of calcium source and the filtration and verification confirmed that the technology of pre-addition on S. pasteurii, static nucleation, undisturbed filtration, and shade drying could produce the highest yield of CaCO3, and also could produce the highest yield of calcite. 

Comments 5:

Bacteria names should be written in italic, the authors should check the whole manuscript and correct where needed.

Response 5:

We apologize for this error. We have carefully reviewed the entire manuscript and corrected all bacterial names to be properly italicized. Except for the abstract, all the strain names of “Sporosarcina pasteurii” in this paper were abbreviated as " S. pasteurii".

Correction 5:

Line 103

In this paper, Sporosarcina pasteurii (abbreviated hereafter as S. pasteurii) produced by DSMZ of Leibniz Institute in Germany, numbered DSM 33, is selected as the MICP target strain.

In the whole paper

……S. pasteurii……

Reviewer 3 Report

Comments and Suggestions for Authors

Although the topic of the paper "Research on MICP restoration technology for earthen city wall damaged by primary vegetation capping in China" is current from the aspects of MICP and the protection of earthen sites in China that represents a world-class challenge, the paper has limitations and shortcomings that may not be suitable for publication in this journal.

First of all, the paper was not written in accordance with Instructions for Authors, when it comes to citing literature and the reference description. Second, the abbreviation of the bacterial name cannot be Pasteurii; maybe S. pasteurii! Furthermore, the abbreviation should not be a title of the chapter headings (e.g. 3.1. FIE). Also:

  • The experimental design (Table 2) is poorly presented, and it is unclear which values were actually tested together or how many replicates were run per condition.
  • The Methods language is often conditional (“should be recorded/planned”) rather than stating actual practice, obscuring whether procedures were followed and replicated.
  • The RF-RFE-CV analysis is built on just 111 data points, which is small given the many factors (media, pH, DO, etc.).
  • There are contradictions and unexplained steps. For example, Table 8 shows centrifugation yields a higher calcite fraction than static filtration, yet the conclusion recommends “undisturbed filtration”.
  • The authors mentioned Pasteurii cultures and mixtures through paper, but they used only one strain from DSM collection.

Author Response

Comments 1:

The paper was not written in accordance with Instructions for Authors, when it comes to citing literature and the reference description.

Response 1:

We apologize for this oversight. We have reformatted the reference list and the reference description according to the journal's guidelines, numbering all references in the order of their citation in the text. 

Comments 2:

The abbreviation of the bacterial name cannot be Pasteurii; maybe S. pasteurii!

Response 2:

We apologize for the error. Except for the abstract, all the strain names of “Sporosarcina pasteurii” in this paper were abbreviated as " S. pasteurii".

Correction 2:

Line 103

In this paper, Sporosarcina pasteurii (abbreviated hereafter as S. pasteurii) produced by DSMZ of Leibniz Institute in Germany, numbered DSM 33, is selected as the MICP target strain.

In the whole paper

……S. pasteurii……

Comments 3:

The abbreviation should not be a title of the chapter headings (e.g. 3.1. FIE).

Response 3:

We apologize for the error. The title of the chapter headings has been changed into full names.

Correction 3:

  1. Results

3.1. Feature importance evaluation

3.1.1. Longitudinal analysis

3.1.2. Cross-sectional analysis

3.2. Breeding conditions

3.2.1. Media

3.2.2. Temperature

3.2.3. pH value

3.2.4. Dissolved oxygen

3.2.5. Inoculation age

3.2.6. Inoculation volume

Comments 4:

The experimental design (Table 2) is poorly presented, and it is unclear which values were actually tested together or how many replicates were run per condition.

Response 4:

We apologize for the error. Table 2 was redrawn.

Comments 5:

The Methods language is often conditional (“should be recorded/planned”) rather than stating actual practice, obscuring whether procedures were followed and replicated.

Response 5:

We apologize for this error. We have revised the entire Materials and methods section to consistently use passive voice, as recommended for academic writing in this discipline.

Correction 5:

Line 177-188

The experimental design for the factors influencing the physiological characteristics of S. pasteurii was shown in Table 2. The experiment on the mineralization of CaCO3 was shown in Table 3. For the factor experiments, multiple samplings at different time points was conducted according to the experimental plan, and the measurement frequency also was planned. Parameters such as culture temperature, pH value, OD600 value, and the difference in electrical conductivity of S. pasteurii liquid between before and after 5-minute intervals were recorded. The remaining bacterial liquid in the conical flask during the culture process was kept under the original conditions for continued cultivation. Calcium acetate was chosen as the calcium source and was provided in solution form. S. pasteurii liquid with high urease was prepared for standby. According to the experimental requirements, the appropriate liquid mixing method and mineralization method was selected. The precision of the weighing balance was 0.0001g.

Comments 6:

The RF-RFE-CV analysis is built on just 111 data points, which is small given the many factors (media, pH, DO, etc.).

Response 6:

Thank you for your valuable suggestions. Due to the constraints of experimental conditions and time, it is difficult for us to conduct supplementary experiments in this revision. Although this study was based on only 111 sets of experimental conditions, the total amount of raw data reached the million-level, meeting the basic requirements of big data analysis.

However, we have discussed the limitations of the study in the discussion section and highlighted the contribution of this paper.

Your suggestions will be an important part of our future work. Thank you again for your help with the manuscript.

Correction 6:

Line 540-545

Although this study was based on only 111 sets of experimental conditions, the total amount of raw data reached the million-level, meeting the basic requirements of big data analysis. The core contribution of this paper lay in proposing and validating a Random Forest methodological framework to provide references for subsequent research to achieve more accurate and interpretable analysis and prediction of mineralization mechanisms.

Comments 7:

There are contradictions and unexplained steps. For example, Table 8 shows centrifugation yields a higher calcite fraction than static filtration, yet the conclusion recommends “undisturbed filtration”.

Response 7:

We apologize for this oversight. I hope that you can understand the reason for choosing “undisturbed filtration” by revising the entire discussion and standardizing term expression on mineralization.

Correction 7:

Line 249-269

2.3. Characterization parameters

 (3) Mineralization characteristics

The yield of mineralization products was measured by the index parameter method. The actual production of CaCO3 was measured on-site. The yield of CaCO3 in output was characterized by the ratio of the actual amount of CaCO3 to the theoretical production of CaCO3, denoted by η, as shown in Formula (6). The formula for calculating the theoretical production of CaCO3 was shown in Formula (7). The yield of calcite in CaCO3 was characterized by the ratio the actual amount of calcite in CaCO3 to the actual amount of CaCO3, denoted by ηcal, as shown in Formula (8).

                     (6)

(7)

                     (8)

where: η is the yield of CaCO3 in output, ηcal is the yield of calcite in CaCO3, in %; mc is the actual amount of CaCO3, mp is the mass of filter paper with sediment attached, mf is the mass of filter paper, m0 is the theoretical amount of CaCO3, m0-a is the theoretical amount of CaCO3 based on the mass of calcium source, m0-u is the theoretical amount of CaCO3 based on the mass of urea, ma is the mass of calcium source added before precipitation, mu is the mass of urea added before precipitation, mcal is the mass of calcite in CaCO3, mara is the mass of aragonite in CaCO3, mvat is the mass of vaterite in CaCO3, in g; Ma is the relative molecular mass of calcium source, 158.17, Mc is the relative molecular mass of CaCO3, 100.09, Mu is the relative molecular mass of urea, 60.06, in g.

line 696-767

4.2. S. pasteurii features of mineralization

 (1) The yield of CaCO3 in output

The research on Ca2+ concentration confirmed that the yield of CaCO3 in output was higher when the same low concentration of calcium source and urea solution were used. Xiao (2004) [56] and Ghosh et al. (2005) [57] found that an excessively high enzymatic hydrolysis rate was not conducive to enhanced mineralization, and a slow and stable rate of calcite mineralization deposition was more beneficial for the enhancement effect of carbonate mineralization bacteria. Jamal et al. (2023) [58] and Xu et al. (2024) [59] indicated that this broad adaptability couldn’t be directly translated into the optimal efficiency of the MICP process. Blindly pursuing high urease activity might be counterproductive. When the urea solution concentration was 1M, the net mass and the yield of CaCO3 in output were higher when the Ca2+ concentration was 1.3M. It can be seen that when the urease activity of S. pasteurii was constant, the ratio of urea to Ca2+ concentration at 1M:1.3M was more conducive to the formation of high-quality net mass and the yield of CaCO3 in output.

As shown in table 8, with the filtration methods, centrifugal filtration was faster than undisturbed filtration, but the yield of CaCO3 in output followed undisturbed filtration > centrifugal filtration, with an error of 5.2% to 5.4%. With the drying method, oven drying was quicker than room temperature drying in the shade, but the yield of CaCO3 in output followed room temperature drying in the shade > oven drying, with an error of 3% to 3.1%. However, based on the fact that centrifugal filtration could quickly and thoroughly collect all solid precipitates onto the filter paper, in many cases, considering the time-saving aspect, centrifugal filtration and oven drying were often adopted to speed up the process, with an error of about 8.3%.

Therefore, to maximize the yield of CaCO3 in output, it was recommended to follow the condition of the ratio of urea to Ca2+ concentration at 1M:1.3M, undisturbed filtration and room temperature drying in the shade.

(2) The yield of calcite in CaCO3

As shown in Figure 11 (b) - (d), the crystals of the mineralization products in group B were mostly needle-like, while those in group D were stacked to form disc-shaped structures ranging from 10 to 20μm. Group C presented granular and blocky forms, with some crystals bonded together, and the crystal size ranged from 2 to 15μm. This indicated that the addition order of S. pasteurii had an impact on the appearance of the mineralization products. By comparing the data of Group B, Group C and Group D in Table 7 and the samples in Figure 11 (b) - (d) and Figure 9 (e), different addition methods of S. pasteurii had different impacts on the crystal form of the mineralization product. The premix method was most conducive to the yield of calcite in CaCO3, which could reach 100%.

As shown in the relevant data of Group I, Group II, Group III, and Group IV in Table 7, the form of centrifugal filtration and room temperature drying in the shade was more likely to form calcite. Among the filtration methods, the yield of calcite in CaCO3 followed the rules of centrifugal filtration > undisturbed filtration, and the error range was from 0.9% to 1.1%. Among the drying methods, the yield of calcite in CaCO3 followed the rules of room temperature drying in the shade > oven drying, and the error range was from 1.8% to 2%.

Therefore, to maximize the yield of calcite in CaCO3, it was recommended to follow the condition of premix method of S. pasteurii, quiescent reaction, centrifugal filtration, and room temperature drying in the shade.

(3) Comprehensive analysis

A comparison of the recommended conditions for the yield of CaCO3 in output and the yield of calcite in CaCO3 revealed that the conflicting condition was the filtration method. The yield of CaCO3 in output needed undisturbed filtration, while the yield of calcite in CaCO3 needed centrifugal filtration.

The lower yield of CaCO₃ in output observed in centrifugal filtration may be attributed to the loss of tiny particles during the post-centrifugation process. The high gravity during centrifugation effectively precipitated larger and well-crystallized particles. However, it also caused some tiny CaCO₃ precursors to be lost along with the supernatant when poured on the filter paper. The higher yield of calcite in CaCO₃ observed in centrifugal filtration may be attributed to the fact that calcite particles were larger in size compared to the particles of aragonite and vaterite, making them more likely to remain on the filter paper.

Among the mineralization products obtained from Group I, II, III and IV, the yield of calcite in outpt were 4.27%, 4.98%, 4.15%, and 2.65%, respectively. Under the same condition of room temperature drying in the shade, the difference in calcite content between undisturbed filtration in Group I and centrifugal filtration in Group II was only 0.71%, indicating no significant difference in crystal form selectivity between them. Meanwhile, the yield of CaCO₃ in output with the condition of undisturbed filtration was higher than that of centrifugal filtration in 5.2%–5.4%, demonstrating a clear advantage in output. Therefore, considering the stability of output, the condition of undisturbed filtration was more appropriate.

In summary, to maximize the yield of CaCO3 in output and the yield of calcite in CaCO3, it was recommended to follow the condition of the ratio of urea to Ca2+ concentration at 1M:1.3M, premix method of S. pasteurii, quiescent reaction, undisturbed filtration, and room temperature drying in the shade.

Comments 8:

The authors mentioned Pasteurii cultures and mixtures through paper, but they used only one strain from DSM collection.

Response 8:

We apologize for this mistake in language expression. The S. pasteurii was merely a strain, and the whole research in this paper was conducted on various cultures and mixtures of this strain.

Correction 8:

Line 271-274

Based on the fact that the S. pasteurii strain exhibited significant individual variability, leading to considerable variation in the results of repeated experiments. Moreover, the influence of each feature on the prediction target for the S. pasteurii strain varied greatly, and there was a high likelihood of multi-collinearity among the features.

Reviewer 4 Report

Comments and Suggestions for Authors

General comment

This study investigates various factors influencing the mineralization activity of Sporosarcina pasteurii. The authors have identified optimal conditions for urease activity and calcite production, with the aim of applying microbial induced calcite precipitation (MICP) in root-soil composites. The work aligns well with the journal’s aims and scopes, and the results are suitable for publication. However, the innovative aspects of the work should be further clarified and emphasized based on the comments provided.

Reviews required

  1. Lines 65-68. A significant application of MICP relates to concrete. This should be incorporated into the introduction, supported by the following additional references:
    1. https://doi.org/10.1016/j.conbuildmat.2019.02.178.
    2. https://doi.org/10.1061/JMCEE7.MTENG-19564.
  2. At the end of the introduction, briefly outline the scope of the study, including the main objectives and types of investigations conducted.
  3. Section 2, lines 131–132: Would it be possible to include a discussion of the substrate of interest? Clarify the specific substrate or environmental conditions relevant to the study.
  4. Throughout the manuscript, I recommend using the present simple tense when referring to tables and figures for consistency and clarity.
  5. Section 3: Report more quantified results, such as percentage reductions or increases, to facilitate clearer understanding of the effects observed.
  6. Table 4: Although the text states that X, U, and u are reported in this table, only X and U are visible. Please clarify or include the missing data.
  7. Section 4: This section should discuss the potential practical implications of the findings, including how the results can inform real-world applications and guide future implementation.

Author Response

Comments 1:

Lines 65-68. A significant application of MICP relates to concrete. This should be incorporated into the introduction, supported by the following additional references: 

Response 1:

Thank you for your valuable suggestions. I listed the additional references above metioned.

Comments 2:

At the end of the introduction, briefly outline the scope of the study, including the main objectives and types of investigations conducted.

Response 2:

Thank you for your valuable suggestions. I have added a paragraph to briefly outline the scope of the study.

Correction 2:

Line 131-135

This study designed a research plan that combined multi-factor experiments with mathematical models. The main objective was to systematically reveal the interaction of multiple factors on characteristics of growth, urease production, and mineralization in MICP process and their importance weights, and to construct a comprehensive structural understanding of the mineralization mechanism of S. pasteurii.

Comments 3:

Section 2, lines 131–132: Would it be possible to include a discussion of the substrate of interest? Clarify the specific substrate or environmental conditions relevant to the study.

Response 3:

Thank you for your valuable suggestions. I listed more details of the substrate.

Correction 3:

Line 138-157

The sampling site of Jinyang Ancient City is located in Jinyuan District, southwest of Taiyuan City, Shanxi Province, China. It includes exposed parts above ground and buried parts underground. The existing exposed part has a trapezoidal profile with distinct rammed layers, with a total residual length of approximately 800 m. The height is 3-8 m. The width of the bottom is about 15 m, and that of the top is 4-10 m. The vegetation distribution on the city wall was uneven overall. The western section of the wall (Figure 2) supported a mix of shrubs and small trees, while the eastern section was mainly covered with herbs and vines. Those plants had shallow root system, mostly ranging from 200 mm to 300 mm, with weak main roots and dense fibrous roots, forming root-soil composites. Shang et al. (2024 (a)) [56] and Shang et al. (2024 (b)) [57] indicated the soil analysis from the site in Jinyang Ancient City that the predominant soil type was sandy loam, classified as non-collapsible loess. With increasing soil depth, the moisture content showed a corresponding rise. The primary soil salts consisted of CaSO4 and MgSO4, with pH values ranging from 7.1 to 7.45 and a specific gravity between 2.69 and 2.7. The volume moisture content of dry soil was about 0.132m3·m-3, and the volume moisture content of wet soil was about 0.283m3·m-3. The sampling date was May 15, 2022. We didn’t obtain the permission document, but in 2023, we were in charge of Shanxi Provincial Cultural Relics Technology Program of China (2023KT15) organized by the Shanxi Provincial Cultural Relics Bureau, which clearly stipulated that the research scope was Western City wall in Jinyang Ancient City.

Comments 4:

Throughout the manuscript, I recommend using the present simple tense when referring to tables and figures for consistency and clarity.

Response 4:

We apologize for this oversight. I revised the tense as into the present simple tense when referring to tables and figures for consistency and clarity.

Correction 4:

Line 173-176: Table 1.

There is no urea in the official medium. Wei (2023) [53] confirmed that the addition of urea to the medium could significantly stimulate S. pasteurii to produce more urease, thereby increasing the mineralization yield by S. pasteurii. **“—" indicates that the factor is empty.Gelling agent is used only in a solid medium and does not need to be added to a liquid medium.

Line 191-200: Table 2.

Experiment names explanation: M refers to the three different culture media. T refers to the culture temperature of media. pH refers to the pH value. DO refers to dissolved oxygen. IA refers to inoculation age of S. pasteurii. IV refers to inoculation volume of S. pasteurii. t refers to the cultivation time of mixture. G refers to the cultivation generation of S. pasteurii.

Symbol explanation: “/” indicates a single-factor element, and “—" indicates that the factor is not measured.

Experimental group description: The optimization design involved a total of 115 experiments, among which 111 sets of data are valid and 4 sets were invalid. The invalid data are as follows: M—M2 (1 invalid data), M—M3 (1 invalid data), T—M1 (1 invalid data), and T—M2 (1 invalid data).

Line 202-218: Table 3.

Addition represents addition order of S. pasteurii. When mixing the liquid, a KQ250DE numerically controlled ultrasonic cleaner is used to shake for 2-3 minutes to ensure the liquid is well mixed. Premix represents to premix the liquid of S. pasteurii and urea, and then to add Ca2+ concentration. Co-add represents to mix the liquid of S. pasteurii, urea, and Ca2+ concentration together. Post-add represents to premix Ca2+ concentration and urea, and then to add the liquid of S. pasteurii.

Reaction represents the reaction condition of S. pasteurii, urea, and Ca2+ concentration. Shaking represents the mixture reaction with orbital shaking condition. The culture temperature was set to 30℃ and oscillation speed was set to 170r/rpm for 24 hours. Quiescent represents the mixture reaction without orbital shaking condition.

Filtration represents the filtration method for output. Centifuge represents the filtration method with centrifugation. For centrifugation, the HC-3018 centrifuge is used into 3 minutes. It can be used again when the sample is still turbid, until the solid and liquid is completely separated. Undisturbed represents the filtration method without centrifugation, leaving mixture undisturbed.

Drying represents the drying method for output. Oven represents the low-temperature drying method with oven. For low-temperature drying, the oven temperature is set to 30℃ in 24 hours. Room represents the room temperature drying method in the shade without oven.

Line 316-318: Table 4.

B refers to the batch of S. pasteurii. SD refers to standard deviation.

The variables X and U that might cause causal confusion. Therefore, when considering u, both of them were excluded.

Line 426: Table 6.

“—" indicates that no experiment is conducted or no conclusion is drawn.

Line 514: Table 7.

Calcite yield represents the yield of calcite in CaCO3.

Line 532: Table 8.

CaCO3 yield represents the yield of CaCO3 in output.

Comments 5:

Section 3: Report more quantified results, such as percentage reductions or increases, to facilitate clearer understanding of the effects observed.

Response 5:

Thank you for your valuable suggestions. I added more quantified results in section 4 to facilitate clearer support for discussion. I hope that you can understand the conclusions by revising the entire discussion and standardizing term expression on mineralization.

Correction 5:

4.2. S. pasteurii features of mineralization

 (1) The yield of CaCO3 in output

The research on Ca2+ concentration confirmed that the yield of CaCO3 in output was higher when the same low concentration of calcium source and urea solution were used. Xiao (2004) [56] and Ghosh et al. (2005) [57] found that an excessively high enzymatic hydrolysis rate was not conducive to enhanced mineralization, and a slow and stable rate of calcite mineralization deposition was more beneficial for the enhancement effect of carbonate mineralization bacteria. Jamal et al. (2023) [58] and Xu et al. (2024) [59] indicated that this broad adaptability couldn’t be directly translated into the optimal efficiency of the MICP process. Blindly pursuing high urease activity might be counterproductive. When the urea solution concentration was 1M, the net mass and the yield of CaCO3 in output were higher when the Ca2+ concentration was 1.3M. It can be seen that when the urease activity of S. pasteurii was constant, the ratio of urea to Ca2+ concentration at 1M:1.3M was more conducive to the formation of high-quality net mass and the yield of CaCO3 in output.

As shown in table 8, with the filtration methods, centrifugal filtration was faster than undisturbed filtration, but the yield of CaCO3 in output followed undisturbed filtration > centrifugal filtration, with an error of 5.2% to 5.4%. With the drying method, oven drying was quicker than room temperature drying in the shade, but the yield of CaCO3 in output followed room temperature drying in the shade > oven drying, with an error of 3% to 3.1%. However, based on the fact that centrifugal filtration could quickly and thoroughly collect all solid precipitates onto the filter paper, in many cases, considering the time-saving aspect, centrifugal filtration and oven drying were often adopted to speed up the process, with an error of about 8.3%.

Therefore, to maximize the yield of CaCO3 in output, it was recommended to follow the condition of the ratio of urea to Ca2+ concentration at 1M:1.3M, undisturbed filtration and room temperature drying in the shade.

(2) The yield of calcite in CaCO3

As shown in Figure 11 (b) - (d), the crystals of the mineralization products in group B were mostly needle-like, while those in group D were stacked to form disc-shaped structures ranging from 10 to 20μm. Group C presented granular and blocky forms, with some crystals bonded together, and the crystal size ranged from 2 to 15μm. This indicated that the addition order of S. pasteurii had an impact on the appearance of the mineralization products. By comparing the data of Group B, Group C and Group D in Table 7 and the samples in Figure 11 (b) - (d) and Figure 9 (e), different addition methods of S. pasteurii had different impacts on the crystal form of the mineralization product. The premix method was most conducive to the yield of calcite in CaCO3, which could reach 100%.

As shown in the relevant data of Group I, Group II, Group III, and Group IV in Table 7, the form of centrifugal filtration and room temperature drying in the shade was more likely to form calcite. Among the filtration methods, the yield of calcite in CaCO3 followed the rules of centrifugal filtration > undisturbed filtration, and the error range was from 0.9% to 1.1%. Among the drying methods, the yield of calcite in CaCO3 followed the rules of room temperature drying in the shade > oven drying, and the error range was from 1.8% to 2%.

Therefore, to maximize the yield of calcite in CaCO3, it was recommended to follow the condition of premix method of S. pasteurii, quiescent reaction, centrifugal filtration, and room temperature drying in the shade.

(3) Comprehensive analysis

A comparison of the recommended conditions for the yield of CaCO3 in output and the yield of calcite in CaCO3 revealed that the conflicting condition was the filtration method. The yield of CaCO3 in output needed undisturbed filtration, while the yield of calcite in CaCO3 needed centrifugal filtration.

The lower yield of CaCO₃ in output observed in centrifugal filtration may be attributed to the loss of tiny particles during the post-centrifugation process. The high gravity during centrifugation effectively precipitated larger and well-crystallized particles. However, it also caused some tiny CaCO₃ precursors to be lost along with the supernatant when poured on the filter paper. The higher yield of calcite in CaCO₃ observed in centrifugal filtration may be attributed to the fact that calcite particles were larger in size compared to the particles of aragonite and vaterite, making them more likely to remain on the filter paper.

Among the mineralization products obtained from Group I, II, III and IV, the yield of calcite in outpt were 4.27%, 4.98%, 4.15%, and 2.65%, respectively. Under the same condition of room temperature drying in the shade, the difference in calcite content between undisturbed filtration in Group I and centrifugal filtration in Group II was only 0.71%, indicating no significant difference in crystal form selectivity between them. Meanwhile, the yield of CaCO₃ in output with the condition of undisturbed filtration was higher than that of centrifugal filtration in 5.2%–5.4%, demonstrating a clear advantage in output. Therefore, considering the stability of output, the condition of undisturbed filtration was more appropriate.

In summary, to maximize the yield of CaCO3 in output and the yield of calcite in CaCO3, it was recommended to follow the condition of the ratio of urea to Ca2+ concentration at 1M:1.3M, premix method of S. pasteurii, quiescent reaction, undisturbed filtration, and room temperature drying in the shade.

Comments 6:

Table 4: Although the text states that X, U, and u are reported in this table, only X and U are visible. Please clarify or include the missing data.

Response 6:

I apologize for the previous mistake. At that time, the data related to "u" was deleted considering the layout aesthetics. Now, the data for "u" has been filled in.

Comments 7:

Section 4: This section should discuss the potential practical implications of the findings, including how the results can inform real-world applications and guide future implementation.

Response 7:

Thank you for your valuable suggestions. I have added this part of the content to the last paragraph.

Correction 7:

  1. Discussion

Although this study was based on only 111 sets of experimental conditions, the total amount of raw data reached the million-level, meeting the basic requirements of big data analysis. The core contribution of this paper lay in proposing and validating a Random Forest methodological framework to provide references for subsequent research to achieve more accurate and interpretable analysis and prediction of mineralization mechanisms.

4.1. S. pasteurii features of urease production

(3) Comprehensive analysis

It was known that U was a core functional indicator for predicting and evaluating the mineralization efficiency of S. pasteurii. Longitudinal analysis (Table 4) showed that when the characteristic parameters of U interacted with X, the number of characteristic parameters reduced from 8 to 6, and X (1.016) became a key characteristic of S. pasteurii, which was far more important than other characteristic parameters. Ignoring the continuity of t, the cross-sectional data (Table 5) of S. pasteurii showed that the weights of X-4h and X-0h at 4-hour were 0.589 and 0.263 respectively, the weight of X-8h at 8-hour was 0.242, the weight of X-12h at 12-hour was 0.248, and the weights of X-0h and X-24h at 24-hour were 1.127 and 0.259 respectively. X was high importance feature at multiple time points such as at 4-hour, 8-hour, 12-hour and 24-hour, and even at 24-hour, X-0h became a key feature. That was, the best response time was at around 24-hour, which proved that urease activity was highly dependent on the initial X in the later stage of culture, and the growth potential in the early stage was fully reflected at this stage. In other words, X was a key driving factor for characterizing U, and its inclusion in U prediction model could reduce the number of characteristic parameters and improve the explanatory power of each characteristic parameter. Its importance was confirmed in both longitudinal and cross-sectional data analysis. It can be seen that U was closely related to the initial pH value and the number of bacteria. A higher initial X could prompt S. pasteurii to enter the rapid proliferation period faster, which would promote the overall increase of X and U. The above conclusion was in line with the general law of bacterial growth.

Longitudinal analysis (Table 4) indicated that t (1.517) was the most critical factor influencing X, with the ranking of highly influential factors followed the rule of IV (0.255) > M (0.21) > IA (0.208) > pH (0.173). Cross-sectional analysis revealed distinct temporal patterns in feature importance. pH-4h and pH-0h at 4-hour exhibited weights of 1.055 and 0.189, respectively, while X-0h had a weight of 0.154. pH-0h value at 8-hour decreased to 0.498, whereas X-0h increased significantly to 1.477. pH-0h and pH-12h at 12-hour were weighted at 0.42 and 0.116, respectively, with X-0h remaining high at 1.317. The influence of pH and X-0h at 24-hour diminished, while IA and G emerged as dominant factors, with weights of 0.486 and 0.322, respectively. Redundancy analysis of bacterial concentration-related features demonstrated a redundancy weight of 1.052 between t and pH, and 0.917 between IA and IV, suggesting potential multi-collinearity issues. In the growth curve of S. pasteurii, pH was identified as the key determinant during the lag phase, with pH-4h and pH-0h at 4-hour weighted at 1.055 and 0.189, respectively—indicating strong regulation by the initial culture medium environment. The results of pH experiment demonstrated that maintaining pH within the range of 7.25 - 9 could effectively shorten the duration of the lag phase, further validating the decisive influence of pH on X during the lag phase. During the exponential growth phase, X-0h at 8-hour and 12-hour became the predominant factor, with weights of 1.477 and 1.317, aligning well with the expected growth kinetics of S. pasteurii during logarithmic proliferation. This highlighted the decisive impact of inoculation strategy on U synthesis in this phase. In contrast, during the stationary or deceleration phase (24 h), influencing factors became more diverse, with weights of 0.486 and 0.322 on IV and G increasing in relative importance, reflecting a complex regulatory stage of bacterial growth. This shift was likely attributable to nutrient depletion and the accumulation of metabolic by-products in later culture stages, which collectively suppress growth rates. These findings were in close agreement with the established growth profile of S. pasteurii under nutrient-limited conditions and corroborate previous reports by Wang (2009) [23] and Wei (2023) [53], thereby enhancing the reliability and generalizability of the conclusions.

In addition to X, the ranking of high-importance features for U in the longitudinal analysis followed the rules of t (0.236) > IA (0.213) > B (0.177). In the cross-sectional analysis, U exhibited pH-4h at 4-hour showing a weight of 0.532. pH-0h at 8-hour reached the highest observed weight of 1.566. pH-0h at 12-hour registered 0.32 and pH-24h at 24-hour contributed a weight of 0.109. These results indicated that pH was a key determinant of S. pasteurii in U, particularly during the early exponential growth phase. The peak importance of pH-0h at 8-hour underscored its critical role in modulating U, further validating pH as a major influencing factor in cross-sectional analysis.

u served as a core quality indicator for assessing U, widely utilized in the screening of high-efficiency strains and the optimization of culture conditions. Longitudinal analysis revealed that the key determinants of u were ranked in descending order of culture time (1.341) > inoculation age (1.234), with pH emerging as a highly influential factor (0.916). Cross-sectional analysis demonstrated distinct temporal patterns in regulatory influences. During the early growth phase, u in S. pasteurii was predominantly governed by pH conditions. pH-0h exhibited a consistent weight of 0.580 at 4-hour and 8-hour, while pH-4h and pH-8h registered weights of 0.316 and 0.282, respectively. As cultivation progresses at 12-hour and 24-hour, the influence of IV increased markedly, with importance weights of 0.882 and 0.355. Concurrently, the role of pH diminished—pH-0h at 12-hour dropped to 0.112, and pH-24h and pH-0h at 24-hour contributed only 0.280 and 0.147, respectively. Redundancy analysis of u revealed a redundancy weight of 0.956 between IA and IV, suggesting a significant degree of multi-collinearity. Cross-sectional evaluation identified pH and IV as primary high-importance features. The significance of pH was consistently validated across multiple time points, whereas the impact of IV became prominent after 12-hour, highlighting its growing regulatory role in the mid-to-late culture phase.

4.2. S. pasteurii features of mineralization

 (1) The yield of CaCO3 in output

The research on Ca2+ concentration confirmed that the yield of CaCO3 in output was higher when the same low concentration of calcium source and urea solution were used. Ghosh et al. (2005) [59] found that an excessively high enzymatic hydrolysis rate was not conducive to enhanced mineralization, and a slow and stable rate of calcite mineralization deposition was more beneficial for the enhancement effect of carbonate mineralization bacteria. Jamal et al. (2023) [60] and Xu et al. (2024) [61] indicated that this broad adaptability couldn’t be directly translated into the optimal efficiency of the MICP process. Blindly pursuing high urease activity might be counterproductive. When the urea solution concentration was 1M, the net mass and the yield of CaCO3 in output were higher when the Ca2+ concentration was 1.3M. It can be seen that when the urease activity of S. pasteurii was constant, the ratio of urea to Ca2+ concentration at 1M:1.3M was more conducive to the formation of high-quality net mass and the yield of CaCO3 in output.

As shown in table 8, with the filtration methods, centrifugal filtration was faster than undisturbed filtration, but the yield of CaCO3 in output followed undisturbed filtration > centrifugal filtration, with an error of 5.2% to 5.4%. With the drying method, oven drying was quicker than room temperature drying in the shade, but the yield of CaCO3 in output followed room temperature drying in the shade > oven drying, with an error of 3% to 3.1%. However, based on the fact that centrifugal filtration could quickly and thoroughly collect all solid precipitates onto the filter paper, in many cases, considering the time-saving aspect, centrifugal filtration and oven drying were often adopted to speed up the process, with an error of about 8.3%.

Therefore, to maximize the yield of CaCO3 in output, it was recommended to follow the condition of the ratio of urea to Ca2+ concentration at 1M:1.3M, undisturbed filtration and room temperature drying in the shade.

(2) The yield of calcite in CaCO3

As shown in Figure 12 (b) - (d), the crystals of the mineralization products in group B were mostly needle-like, while those in group D were stacked to form disc-shaped structures ranging from 10 to 20μm. Group C presented granular and blocky forms, with some crystals bonded together, and the crystal size ranged from 2 to 15μm. This indicated that the addition order of S. pasteurii had an impact on the appearance of the mineralization products. By comparing the data of Group B, Group C and Group D in Table 7 and the samples in Figure 12 (b) - (d) and Figure 9 (e), different addition methods of S. pasteurii had different impacts on the crystal form of the mineralization product. The premix method was most conducive to the yield of calcite in CaCO3, which could reach 100%.

As shown in the relevant data of Group I, Group II, Group III, and Group IV in Table 7, the form of centrifugal filtration and room temperature drying in the shade was more likely to form calcite. Among the filtration methods, the yield of calcite in CaCO3 followed the rules of centrifugal filtration > undisturbed filtration, and the error range was from 0.9% to 1.1%. Among the drying methods, the yield of calcite in CaCO3 followed the rules of room temperature drying in the shade > oven drying, and the error range was from 1.8% to 2%.

Therefore, to maximize the yield of calcite in CaCO3, it was recommended to follow the condition of premix method of S. pasteurii, quiescent reaction, centrifugal filtration, and room temperature drying in the shade.

(3) Comprehensive analysis

A comparison of the recommended conditions for the yield of CaCO3 in output and the yield of calcite in CaCO3 revealed that the conflicting condition was the filtration method. The yield of CaCO3 in output needed undisturbed filtration, while the yield of calcite in CaCO3 needed centrifugal filtration.

The lower yield of CaCO₃ in output observed in centrifugal filtration may be attributed to the loss of tiny particles during the post-centrifugation process. The high gravity during centrifugation effectively precipitated larger and well-crystallized particles. However, it also caused some tiny CaCO₃ precursors to be lost along with the supernatant when poured on the filter paper. The higher yield of calcite in CaCO₃ observed in centrifugal filtration may be attributed to the fact that calcite particles were larger in size compared to the particles of aragonite and vaterite, making them more likely to remain on the filter paper.

Among the mineralization products obtained from Group I, II, III and IV, the yield of calcite in outpt were 4.27%, 4.98%, 4.15%, and 2.65%, respectively. Under the same condition of room temperature drying in the shade, the difference in calcite content between undisturbed filtration in Group I and centrifugal filtration in Group II was only 0.71%, indicating no significant difference in crystal form selectivity between them. Meanwhile, the yield of CaCO₃ in output with the condition of undisturbed filtration was higher than that of centrifugal filtration in 5.2%–5.4%, demonstrating a clear advantage in output. Therefore, considering the stability of output, the condition of undisturbed filtration was more appropriate.

In summary, to maximize the yield of CaCO3 in output and the yield of calcite in CaCO3, it was recommended to follow the condition of the ratio of urea to Ca2+ concentration at 1M:1.3M, premix method of S. pasteurii, quiescent reaction, undisturbed filtration, and room temperature drying in the shade.

Reviewer 5 Report

Comments and Suggestions for Authors

The article is well prepared; I have not found any shortcomings in terms of methodology or discussion, and the conclusions are comprehensive.
In my opinion, the article is worthy of publication.

Materials and methods: This part of the article requires supplementation because it does not describe the parameters of wall humidity, the degree of damage or the material from which the wall is made.

Change the abstract to emphasize Pasteur's analysis of high activity.

Author Response

Comments 1:

Materials and methods: This part of the article requires supplementation because it does not describe the parameters of wall humidity, the degree of damage or the material from which the wall is made.

Response 1:

We apologize for this oversight. The parameters of wall humidity, the degree of damage or the material from which the wall was made in introduction.

Correction 1:

Line 138-157

The sampling site of Jinyang Ancient City is located in Jinyuan District, southwest of Taiyuan City, Shanxi Province, China. It includes exposed parts above ground and buried parts underground. The existing exposed part has a trapezoidal profile with distinct rammed layers, with a total residual length of approximately 800 m. The height is 3-8 m. The width of the bottom is about 15 m, and that of the top is 4-10 m. The vegetation distribution on the city wall was uneven overall. The western section of the wall (Figure 2) supported a mix of shrubs and small trees, while the eastern section was mainly covered with herbs and vines. Those plants had shallow root system, mostly ranging from 200 mm to 300 mm, with weak main roots and dense fibrous roots, forming root-soil composites. Soil analysis from the site in Jinyang Ancient City (Shang et al., 2024) indicated that the predominant soil type was sandy loam, classified as non-collapsible loess. With increasing soil depth, the moisture content showed a corresponding rise. The primary soil salts consisted of CaSO4 and MgSO4, with pH values ranging from 7.1 to 7.45 and a specific gravity between 2.69 and 2.7. The volume moisture content of dry soil was about 0.132m3·m-3, and the volume moisture content of wet soil was about 0.283m3·m-3.

Comments 2:

Change the abstract to emphasize Pasteur's analysis of high activity.

Response 2:

S. pasteurii’s analysis of high activity was emphasized in abstract.

Correction 2:

Line 14-36

Abstract: As a typical representative of soft capping, the primary vegetation capping has both protective and destructive effects on earthen city walls. Addressing its detrimental aspects constitutes the central challenge of the project. Based on the integration of MICP technology with plants offered advantages including soil solidification, erosion resistance, and resilience to dry-wet and freeze-thaw cycles, the application of MICP technology to the root-soil composite was proposed as a potential solution. Employing a combined approach of RF-RFE-CV modeling and microscopic imaging on laboratory samples from Western City Wall of the Jinyang Ancient City in Taiyuan, Shanxi Province, China, key factors and characteristics in the mineralization process of Sporosarcina pasteurii was quantified and observed systematically to define the optimal pathway for enhancing urease activity and calcite yield. The conclusions were as follows. Urease activity of S. pasteurii was primarily regulated by three key parameters with bacterial concentration, pH value, and the intensity of urease activity, which required stage-specific dynamic control throughout the growth cycle. Bacterial concentration consistently emerged as a high-importance feature across multiple time points, with peak effectiveness observed at 24-hour (1.127). pH value remained a highly influential parameter across several time points, exhibiting maximum impact at around 8-hour (1.566). With the intensity of urease activity, pH exerted a pronounced influence during the early cultivation stage, whereas inoculation volume gained increasing importance after 12-hour. To achieve maximum urease activity, it was recommended to use CASO AGAR Medium 220 and the following optimized culture conditions: activation culture time of 27-hour, inoculation age of 16-hour, inoculation volume of 1%, culture temperature of 32°C, initial pH of 8, and oscillation speed of 170 r/min. Furthermore, to maximize the yield of CaCO3 in output and the yield of calcite in CaCO3, it was recommended to follow the condition of the ratio of urea to Ca2+ concentration at 1M:1.3M, pre-mix method of Sporosarcina pasteurii, quiescent reaction, undisturbed filtration, and room temperature drying in the shade.

Round 2

Reviewer 2 Report

Comments and Suggestions for Authors

Dear Editor, dear Authors,

I have carefully read the revised paper. The authors have taken into account all the reviewers' comments, provided adequate responses and explanations, and I propose that the paper be accepted in its current form.

Author Response

Thank you for the comments!

Reviewer 3 Report

Comments and Suggestions for Authors

The paper "Research on MICP restoration technology for earthen city wall damaged by primary vegetation capping in China" still has limitations and shortcomings that may not be suitable for publication in this journal, which are provided in an attachment.

Author Response

Comments 1:

When it comes to citing literature it should be like this Lv (2021) [1] through the whole paper.

Correction 1:

Lv (2021) [1], Lu (2017) [2], Riegl (2010) [3], and Brandi (2016) [4] ……

Zhu et al. (2024) [5], Liu et al. (2021) [6], Bu et al. (2022) [7], Zhang et al. (2023) [8], Boruah (2025) [9], and Varnitha et al. (2021) [10] ……

Wu et al. (2022) [11], Liu et al. (2018) [12], Wang et al. (2025) [13], and Whiftin et al. (2007) [14]……

Xu et al. (2013) [15] and Dayana et al. (2025) [16]……

Wang (2009) [17]……

Álvaro et al. (2018) [18]……

Lutfian et al. (2024) [19], Deng et al. (2025) [20], Wang et al. (2024) [21], and Murugan et al. (2021) [22]……

Chen et al. (2021) [23]……

He et al. (2024) [24]……

Liu et al. (2024) [25], Liu et al. (2020) [26], and Wang et al. (2023) [27]……

Liang et al. (2025) [28], Meghdad et al. (2024) [29], and Lopes et al. (2025) [30]……

Zheng et al. (2022) [31]……

Gong et al. (2024) [32] and Zhu et al. (2025) [33]……

Tang et al. (2020) [34] and Wu et al. (2017) [35]……

Kang et al. (2014) [36] and Wong et al. (2015) [37]……

Achal et al. (2009) [38], Amiya et al. (2025) [39], Lv et al. (2025) [40], Wang et al. (2025) [41], Alvarado et al. (2025) [42], Wang et al. (2024) [43], Do et al. (2025) [44], Yang et al. (2025) [45], Qian et al. (2009) [46], Sun (2019) [47], and Wei et al. (2022) [48]……

Meng et al. (2021) [49] and Rui et al. (2024) [50]……

Wei (2023) [51] and Fan (2022) [52]……

Webster et al. (2006) showed that MICP remediation process could eventually form a layer of calcite (CaCO3) with a thickness of 5 to 40 micrometers on the surface [53].

Shang et al. (2024 (a)) [54] and Shang et al. (2024 (b)) [55]……

These findings were in close agreement with the established growth profile of S. pasteurii under nutrient-limited conditions and corroborate previous reports by Wang (2009) [17] and Wei (2023) [51], thereby enhancing the reliability and generalizability of the conclusions.

Xiao (2004) [56] and Ghosh et al. (2005) [57] found that an excessively high enzymatic hydrolysis rate was not conducive to enhanced mineralization, and a slow and stable rate of calcite mineralization deposition was more beneficial for the enhancement effect of carbonate mineralization bacteria. Jamal et al. (2023) [58] and Xu et al. (2024) [59] indicated that this broad adaptability couldn’t be directly translated into the optimal efficiency of the MICP process. Blindly pursuing high urease activity might be counterproductive.

Comments 2:

Lines 17 and 18 - resilience to dry-wet and freeze- thaw cycles.

Correction 2:

Line 17-18:

Based on the integration of MICP technology with plants offered advantages including soil solidification, erosion resistance, and resilience to dry-wet cycles and freeze-thaw cycles, the application of MICP technology to the root-soil composite was proposed as a potential solution.

Comments 3:

Line 117 - creas3ed

Correction 3:

Line 95:

Liang et al. [30], Meghdad et al. [31], and Lopes et al. [32] showed the terms of resistance to erosion, and Zheng et al. [33] indicated that the cohesion and internal friction angle of the root-soil complex increas3ed by approximately 400% and 120% respectively.

Comments 4:

Line 138 - and the reaction process is shown in Formulas (1) – (2)

Correction 4:

Line 111:

Achal et al. [40], Amiya et al. [41], Lv et al. [42], Wang et al. [43], Alvarado et al. [44], Wang et al. [45], Do et al. [46], Yang et al. [47], Qian et al. [48], Sun [49], and Wei et al. [50] concluded the mineralization mechanism of S. pasteurii (Figure 2) and the reaction process was shown in Formulas (1) – (2).

Comments 5:

Line 184 - 2.69 and 2.7 – Note: Generally, add units.

Correction 5:

Line 149:

The primary soil salts consisted of CaSO4 and MgSO4, with pH values ranging from 7.1 to 7.45  (with no unit) and a specific gravity ranging from 2.69 to 2.7 (with no unit).

Comments 6:

Line 186 - We didn’t did not

Correction 6:

Line 151:

We did not obtain the permission document, but in 2023, we were in charge of Shanxi Provincial Cultural Relics Technology Program of China (2023KT15) organized by the Shanxi Provincial Cultural Relics Bureau, which clearly stipulated that the research scope was Western City wall in Jinyang Ancient City.

Comments 7:

In Table 2 - M (h) or t (h)?

Comments 8:

Line 238 - Addition represents addition order of S. pasteurii. NOTE: Rephrase the sentence.

Comments 9:

Lines 238 and 239 - When mixing the liquid, a KQ250DE numerically controlled ultrasonic cleaner

is was used to shake for 2-3 minutes to ensure the liquid is was well mixed.

Correction 9:

Line 200:

When mixing the liquid, a KQ250DE numerically controlled ultrasonic cleaner was used to shake for 2-3 minutes to ensure the liquid was well mixed.

Comments 10:

Line 491 - Figure 89. Schematic diagram of mMineralization process..

Correction 10:

Line 435

Figure 9. Mineralization process by S. pasteurii. 0min (a), 10min (b), 30min (c), 12h (d), Group A - 24h (e), Group B - 24h (f), Group C - 24h (g), Group D - 24h (h), Group A - 48h (i), Group B - 48h (j), Group C - 48h (k), Group D - 48h (l).

Comments 11:

From line 523 onwards, make sure you number the figures correctly and refer to them in the text

(i.e., line 535 - Figure 11 should be Figure 12, line 553 - Figure 12 should be Figure 13, line 557 -

shown in Figure 10 11 12 (b) - (d). line 636 - The XRD patterns was shown in Figure 10 8 9 (e).etc.)..

Correction 11:

Line 479:

Figure 12. Trend charts with U and CaCO3 at different Ca2+ concentrations. Ca2+ concentration and U (a), Net mass and yield of CaCO3 in output (b).

Line 496:

Figure 13. Trend chart in net mass with Ca2+ concentration under different S. pasteurii addition sequences.

Line 499:

The SEM images of the mineralization products of Group B, Group C, and Group D were shown in Figure 10 (b) - (d).

Line 531:

The XRD patterns was shown in Figure 10 (e).

Line 554:

As shown in Figure 7, the peak of U of S. pasteurii occurred earlier than X.

Comments 12:

Line 707 - U didn’t did not exhibit

Correction 12:

Line 586:

At 4-hour, U did not exhibit a single dominant characteristic.

Comments 13:

Lines 709 and 710 - At 8-hour, X became a crucial characteristic for regulating U, and its weight increased significantly to 1.568 NOTE: Check again the value

Correction 13:

Line 588-591:

At 8-hour, pH-0h became a crucial characteristic for regulating U, and its weight increased significantly to 1.566. The high important feature was X-8h, with a weight of 0.242. It was well known that urease, as an enzyme associated with bacterial metabolism, had an activity closely related to X. A higher X-8h can enhance the overall U.

Comments 14:

Line 797 – The u served as

Correction 14:

Line 676

The u served as a core quality indicator for assessing U, widely utilized in the screening of high-efficiency strains and the optimization of culture conditions.

Comments 15:

Line 832 - couldn’t could not be directly

Correction 15:

Line 700

Jamal et al. [60] and Xu et al. [61] indicated that this broad adaptability could not be directly translated into the optimal efficiency of the MICP process.

Comments 16:

Line 838 - As shown in Table 8

Correction 16:

Line 707

As shown in Table 8, with the filtration methods, centrifugal filtration was faster than undisturbed filtration, but the yield of CaCO3 in output followed undisturbed filtration > centrifugal filtration, with an error of 5.2% to 5.4%.

Comments 17:

Line 851 - As shown in Figure 12 10 (b) - (d),

Correction 17:

Line 720

As shown in Figure 10 (b) - (d), the crystals of the mineralization products in group B were mostly needle-like, while those in group D were stacked to form disc-shaped structures ranging from 10 to 20μm.

Comments 18:

Line 858 - Figure 12 (b) - (d) and Figure 9 (e),... NOTE: Check figures numbers in the whole paper

Correction 18:

Line 726

By comparing the data of Group B, Group C and Group D in Table 7 and the samples in Figure 10 (e), different addition methods of S. pasteurii had different impacts on the crystal form of the mineralization product.
